# A very likely weakening of Pacific Walker Circulation in constrained near-future projections

Mingna Wu [1,2], Tianjun Zhou [1,2,3 ✉], Chao Li [4], Hongmei Li[4], Xiaolong Chen [1,3], Bo Wu[1,3], Wenxia Zhang [1] & Lixia Zhang[1,3]

The observational records have shown a strengthening of the Pacific Walker circulation (PWC) since 1979. However, whether the observed change is forced by external forcing or internal variability remains inconclusive, a solid answer to more societal relevantly question of how the PWC will change in the near future is still a challenge. Here we perform a quantitative estimation on the contributions of external forcing and internal variability to the recent observed PWC strengthening using large ensemble simulations from six state-of-the-art Earth system models. We find the phase transition of the Interdecadal Pacific Oscillation (IPO), which is an internal variability mode related to the Pacific, accounts for approximately 63% (~51–72%) of the observed PWC strengthening. Models with sufficient ensemble members can reasonably capture the observed PWC and IPO changes. We further constrain the projection of PWC change by using climate models' credit in reproducing the historical phase of IPO. The result shows a high probability of a weakened PWC in the near future.

[1] State Key Laboratory of Numerical Modeling for Atmospheric Sciences and Geophysical Fluid Dynamics, Institute of Atmospheric Physics, Chinese Academy of Sciences, 100029 Beijing, China. [2] University of Chinese Academy of Sciences, 100049 Beijing, China. [3] CAS Center for Excellence in Tibetan Plateau Earth Sciences, Chinese Academy of Sciences (CAS), Beijing, China. [4] Max Planck Institute for Meteorology, Hamburg, Germany. ✉email: zhoutj@lasg.iap.ac.cn

The Pacific Walker circulation (PWC) is one of the most important modulators in global climate system[1–6]. The recent high-quality observations constrained by satellite data available since 1979 reveal that the Pacific Walker Circulation (PWC) has strengthened over the past several decades[7–16]. The change of the PWC strength, occurring with a La Niña-like cooling in the eastern Pacific and strengthened surface easterly winds[8,17], has been suggested to be responsible for the recent global warming hiatus[10,11,18–20]. Previous studies have attributed the recent PWC strengthening to internal variability[10,15,18,21] such as the Interdecadal Pacific Oscillation (IPO)[10,22] or the Atlantic Multidecadal Oscillation (AMO)[23]. In addition to internal variability, the accelerated Pacific trade winds and strengthened PWC might be partly due to external forcing, such as aerosol forcing[14]. Although several previous studies have qualitatively suggested that the observed PWC strengthening contains both forced responses and internal variations[14,15], their relative contributions from different factors remain unknown. Moreover, forcing from decadal warming trends since 1990s over the tropical Indian Ocean[8,24] and the Atlantic[11,25,26] could induce inter-basin climate interactions to strengthen the PWC, but there is no agreement yet on which ocean plays the dominant role, hampering an explicit attribution of the PWC change.

The future change of PWC in a warmer climate is another bone of contention. One group of the theoretical studies proposed that the greenhouse-gases(GHGs) induced-warming will increase the tropical atmospheric static stability, thus will drive a slowdown of the PWC[27,28]. In the meanwhile, another group of studies argued that the inhomogeneous warming over the tropical Pacific with increased east-west SST gradient will enhance the PWC[29,30]. The climate models projected a weakening of PWC under global warming on the long-term scale[31–35]. However, internal variability may act to obscure the externally forced change of the PWC in the next few decades, which prohibits a reliable near-term projection of the PWC. It remains unclear whether the human-induced PWC change will overwhelm the magnitude of internal variability and whether a reliable PWC projection for the near-future can be obtained.

Previous assessments about the PWC rely heavily on models from the Coupled Model Intercomparison Project (CMIP)[10–13,32,33]. However, it is insufficient to separate the forced and unforced responses from individual CMIP models with small members of realizations. The recent developed large ensemble simulations from a single model provide a powerful tool to address the roles of internal variabilities in attributing and projecting climate changes[36–38]. Forced by the same radiative forcing but starting from slightly perturbed initial conditions, the ensemble mean can be regarded as the externally forced response, whereas the ensemble spread of all realizations can be taken as the effect from internal climate variabilities. The large ensemble simulations also provide a sufficient assemblage of independent samples to reveal the roles of different internal variabilities.

In this work, based on six sets of large ensemble simulations collected from the US CLIVAR Working Group on Large Ensembles[38], we investigate to what extent the recent strengthening of PWC is attributable to internal variability. We demonstrate that the IPO related to the Pacific has played a leading role in regulating the past and near-future PWC changes.

## Results

**Dominant role of internal variability.** Is the recent PWC strengthening a forced response or modulated by internal variability? To address this question, we examined the PWC change based on the sea-level pressure (SLP) anomaly over the equatorial Pacific during recent decades in MPI-GE with 100 members (see

Methods). The externally-forced PWC change, represented by the ensemble mean of all members, shows a slight increasing trend of 12.9 Pa $(36 \text{ year})^{-1}$ during the period of 1980-2015 (Fig. 1a). In contrast to a weak impact of external forcing, the uncertainty caused by internal variability is evidently large (Fig. 1a). Under the same external forcing, 100 ensemble members exhibit diverse PWC trends, ranging from -73.0 to 106.1 Pa $(36 \text{ year})^{-1}$. The PWC in observation shows an averaged increasing trend of 99.9 Pa $(36 \text{ year})^{-1}$ over 1980-2015, which is, however, within the range of member spread of the ensemble simulation of MPI-GE model (Fig. 1a).

To check the relative role of internal variability, we examine the associated spatial distributions of trends in SLP over the Indo-Pacific. The observational HadSLP2 dataset indicates that the SLP decreased over the western tropical Pacific but increased over the eastern tropical Pacific, indicating a strengthened PWC (Fig. 1b). The MPI-GE ensemble mean rarely captures the observed PWC strengthening and the positive zonal gradient of SLP simulated by the ensemble mean is weaker than that in the observation (Fig. 1c). However, a pronounced increased (decreased) SLP gradient in the composites of 10 members shows the largest strengthening (weakening) of PWC (hereinafter referred to as Max10 and Min10; Fig. 1d, e). Hence, the internal variability can superimpose on the externally-forced change and lead to a large spread in reproducing the PWC trends during 1980–2015 among individual realizations.

**IPO modulates the PWC change.** The strength of the PWC is tightly coupled to the east-west SST gradient over the equatorial Pacific through Bjerknes feedback[1]. To identify the dominant internal mode responsible for the inter-member spread of the PWC change during 1980–2015, we examine the associated spatial distributions of trends in SST and low-level winds (Fig. 2). The observed low-level wind change is characterized by enhanced easterlies in the tropical western Pacific, with an accompanied La Niña-like SST pattern linked to IPO (Fig. 2a). The MPI-GE ensemble mean exhibits slightly strengthened trade wind, but its magnitude is far weaker than that in the observation (Fig. 2a, b). The change of trade wind is associated with the increased zonal SST gradient over the tropical Pacific under global warming[30]. In contrast, trends of Max10 members are similar to the observation (Fig. 2a, c), which is consistent with the strengthened PWC in this period (Fig. 1d). We further calculated the SST trend differences between the Max10 and the Min10 members to identify the dominant internal mode (Fig. 2d). We find that the composite differences resemble the IPO-like pattern in both observation[39] and the internal IPO mode of MPI-GE (Supplementary Fig. 1a), implying that IPO is the dominant internal mode in regulating the PWC variabilities. If IPO switches its phase, the associated east-west contrast in equatorial Pacific SST will change hence the PWC strength.

We examine the relationship between trends in PWC index and IPO index (see Methods) to further verify the dominance of the IPO. Among 100 members, the simulated PWC trends are significantly negatively correlated with the IPO trends ($r = -0.69$, $P < 0.01$) (Fig. 2e), which indicates that PWC change are connected to IPO phase transitions. We further analyze the outputs of the 1021-year MPI-GE pre-industrial control simulation and confirm that the above relationship is robust (Supplementary Fig. 1). The positive IPO-phase-like SST anomalies are associated with a weakened Indo-Pacific SLP gradient (Supplementary Fig. 1a, b), and vice versa. A significant correlation coefficient of $-0.55$ ($-0.58$) is seen between PWC index (PWC trend) and IPO index (IPO trend) (Supplementary Fig. 1c, d).

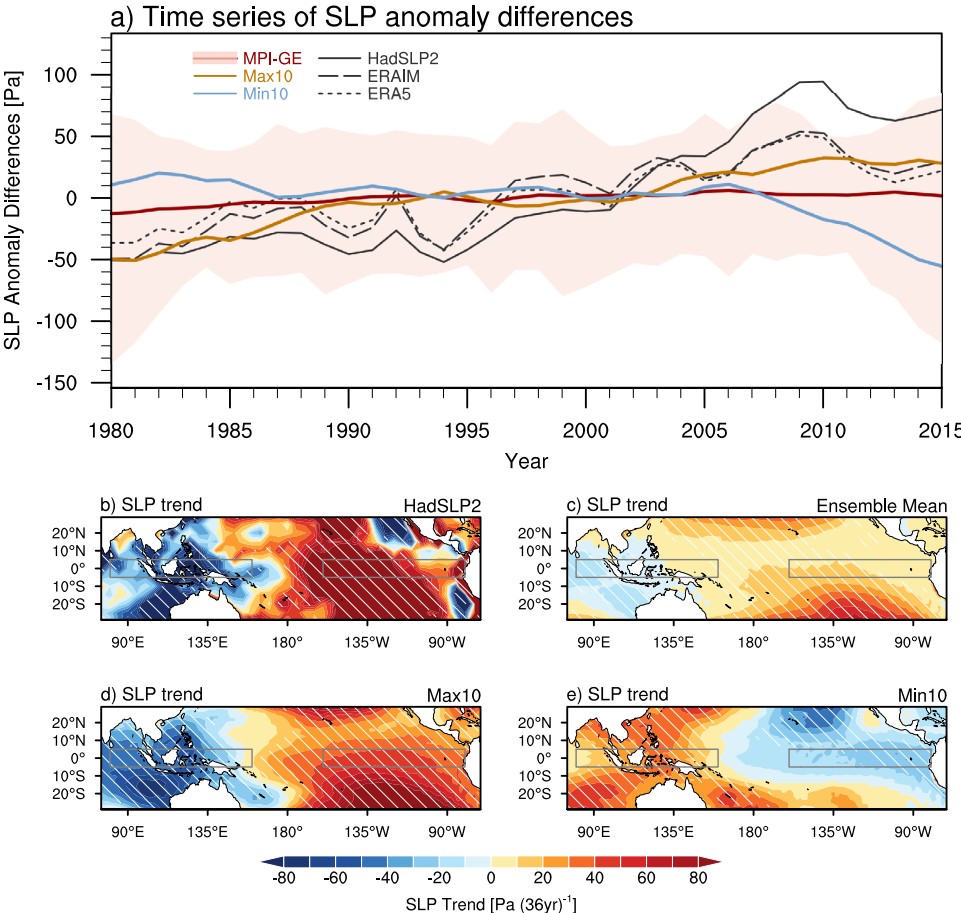

**Fig. 1 Model-simulated temporal evolution and spatial distributions of trends in Sea-level pressure (SLP) during the period of 1980-2015 in MPI-GE.**
**a** Time series of the 9-year running mean of SLP anomaly differences [units: Pa] between equatorial central/eastern Pacific (5°S-5°N, 160°-80°W) and the Indian Ocean/western Pacific (5°S-5°N, 80°-160°E) outlined by rectangles in **b**-**e**. Black lines are derived from the HadSLP2, ERAIM and ERA5 datasets. Red line denotes the ensemble mean of the 100 MPI-GE members. The yellow and blue lines denote 10 ensemble members showing the largest strengthening (Max10) and weakening (Min10) of the PWC during 1980–2015. The uncertainty captured by the red shading is solely due to internal variability. **b**-**e** Spatial distributions of trends in SLP [units: Pa (36 year)$^{-1}$] derived from HadSLP2, MPI-GE ensemble mean, Max10, and Min10, respectively. Slant hatching denotes trends significant at the 95% confidence level.

The AMO may regulate the PWC change through the inter-basin interaction of the North Atlantic and the tropical eastern Pacific SST anomaly, which in turn shifts the Indo-Pacific SST gradient[23]. The observed correlation coefficient between PWC index and AMO index is within the natural variability range (Supplementary Fig. 2e). However, with a large independent sample size, we cannot find a robust coherent relationship between the AMO trend and PWC trend on a 36-year window (Fig. 2f). This indicates that the phase transition of the AMO has limited influence on PWC change during the focused period. Hence it is the IPO that plays the dominant role in modulating the recent PWC strengthening. The observed IPO phase has experienced a shift from positive to negative during the analysis period[40], which has increased the zonal SST gradient over the tropical Pacific to strengthen the PWC.

**Reconciling observation-model discrepancy**. There are various IPO phase evolutions in the MPI-GE members due to random processes in the free runs of the coupled model. To investigate the contribution of the IPO phase transitions to the recent PWC strengthening, we adjust the IPO trends in each MPI-GE member based on the observed IPO change during 1980–2015 (see Methods). We also use the large ensemble simulations (LEs) from other five models to confirm the robustness of the results. We first

exclude the original IPO-related PWC trends, and then add impacts of the observed IPO phase transition to each member through regression (see Methods, Eq. (7)). After adjustment, the ensemble mean of the adjusted ensemble members includes the response to both external forcing and the observed IPO phase transition. The externally forced PWC trends during 1980–2015 range from −21.0 to 9.9 Pa (36 year)$^{-1}$ among the 6 models (Fig. 3a), which is due to the model spread in equatorial zonal SST gradient simulation (Supplementary Fig. 3). After adjustment, the PWC shows an enhanced trend of 57.1 (~30.4–79.0) Pa (36 year)$^{-1}$, which is closer to the observation in magnitude of 99.9 (~67.5–141.7) Pa (36 year)$^{-1}$ (Fig. 3a). The multi-model average of IPO-induced trend is 63.0 (~51.3–72.1) Pa (36 year)$^{-1}$, accounting for approximately 63% (~51–72%) of the observed trend (Fig. 3a). Moreover, all the models converge toward a high probability of a strengthened PWC as the observation after adjustment (Fig. 3b–g). Hence, the phase evolution of IPO is crucial to a successful reproduction of the observed PWC trend.

**Near term PWC projection**. In the near-term projection (2016–2051), the PWC trend shows a large spread among the 270 realizations from six LEs under the RCP8.5 scenario (Supplementary Fig. 4a). The spread can be partly explained by the IPO phase transition, as evinced by the significant negative correlation

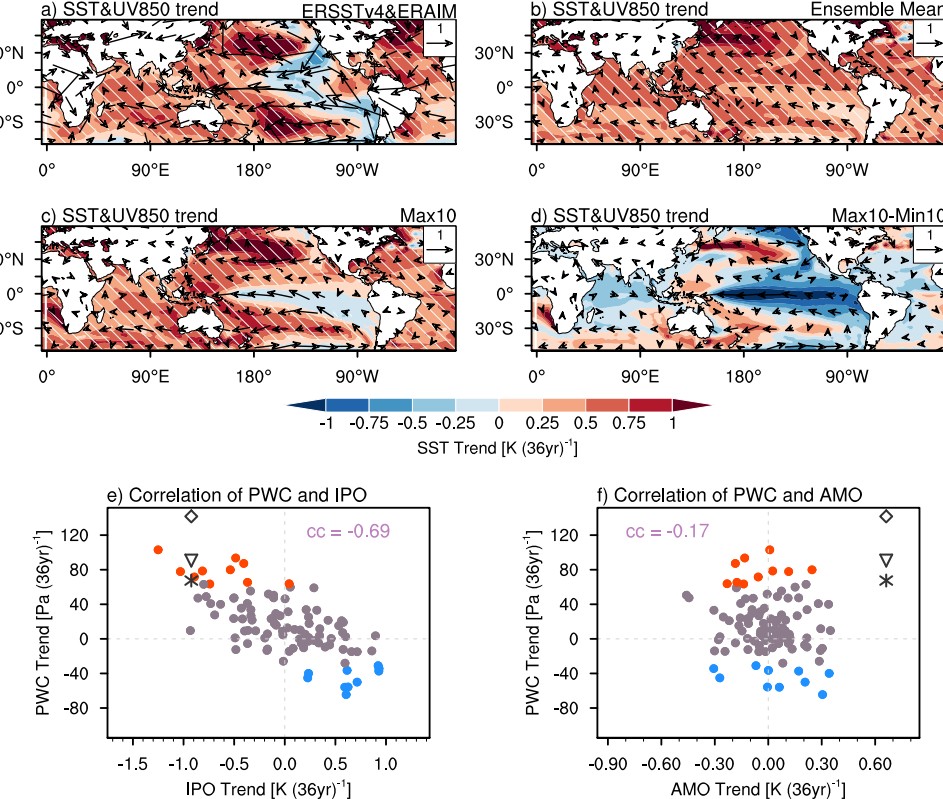

**Fig. 2 Impacts of internal variability on the PWC change during the period of 1980–2015.** Spatial distributions of trends in sea surface temperature [SST; shading, units: K (36 year)$^{-1}$] and wind at 850 hPa [UV850; vectors, units: m s$^{-1}$ (36 year)$^{-1}$] for **a**. ERSST v4 and ERAIM, **b** MPI-GE ensemble mean, **c** Max10, and **d** the Max10-Min10 composite differences. Max10 and Min10 represent 10 ensemble members showing the largest strengthening and weakening of the PWC during 1980-2015, respectively. Slant hatching denotes trends significant at 95% confidence level. **e** Scatter plot of IPO index trends [x axis, units: K (36 year)$^{-1}$] and PWC index trends [y axis, units: Pa (36 year)$^{-1}$] during 1980-2015 among 100 MPI-GE members. Red and blue dots denote the Max10 and Min10 members, respectively. Black diamond, triangle and asterisk denote the observed values from ERSST v4 and HadSLP2, ERSST v4 and ERAIM, and ERSST v4 and ERA5. The correlation coefficient (cc) is indicated at the top of the panel. **f** The same as **e**, but for AMO index trends.

between the projected PWC and IPO trends ($r = -0.51$, $P < 0.01$; Supplementary Fig. 4a). Since the phase of IPO can last for decades, we use IPO to constrain the near-term projection of PWC. We first select the ensemble members with an IPO phase sufficiently close to the observation in the historical simulations (1980–2015) to obtain the near-future (2016–2051) IPO's phase evolution. Among the 270 realizations, we select 13 members whose simulated IPO is closest in sign and magnitude to the observed IPO, with top 5% significant correlations between the member-simulated and observed IPO time series (hereinafter referred to as "BM members"; Supplementary Fig. 4b). Those ensemble members with higher skills in simulating the current IPO are regarded as more reliable in projecting the future IPO. To access the creditability of this method, we choose BM members based on 1950–1985 period to see whether they can reproduce the observed PWC change in 1980–2015. We find that the selected ensemble members which are just in phase with the observed IPO in 1950–1985 do capture the IPO phase shift in 1980–2015 (Supplementary Fig. 5b, c) and performs better than the remaining ensemble members (Supplementary Fig. 5a, b). The weakening of the PWC from the 1950s to 1980s and the strengthening thereafter are well reproduced (Supplementary Fig. 5b, d), suggesting that this approach is creditable. Noted that although the selected BM members simulate a smaller magnitude of the PWC change compared with that in the observation, the observed PWC trend lays within the range of the ensemble member spread, indicating models' ability to reproduce the observed PWC variation (Supplementary Fig. 5a). We then adjust

the IPO trends in all 270 individual members based on the averaged IPO change projected by the BM members (see Methods, Eq. (7)) to quantify the influence of the IPO-related internal variability on the uncertainty in near-term PWC projection.

The IPO trends calculated based on the 36-year running window by BM members change sign from negative to positive, with reversed sign change of the PWC trends for the same period (Fig. 4a). The BM members project an IPO phase change from $-0.90$ K (36 year)$^{-1}$ for 1980–2015 to 0.13 K (36 year)$^{-1}$ for 2016–2051, which indicates a shift toward positive phase of IPO (Fig. 4b). Correspondingly, we find the projected PWC trend by the BM members for the same period is negative with an average rate of $-21.5$ Pa (36 year)$^{-1}$, whereas the simulated PWC trend is positive with an average rate of 32.5 Pa (36 year)$^{-1}$ (Fig. 4c). This reveals that the PWC is very likely to weaken in the coming decades. With the same robust negative-to-positive phase shift of IPO projected by BM members, we find the ensemble mean PWC trend decreases from $-14.2$ to $-22.7$ Pa (36 year)$^{-1}$ after applying the IPO constraint method (Supplementary Fig. 4c). Meanwhile, the probability of a weakened PWC increases from 69% to 83% (Supplementary Fig. 4c). This reveals the large impacts from the IPO-related uncertainty in the near-term PWC projection, as noted in the historical change of the PWC.

## Discussion
We conclude that the recent PWC strengthening is a robust consequence of the phase transition of IPO rather than external forcing.

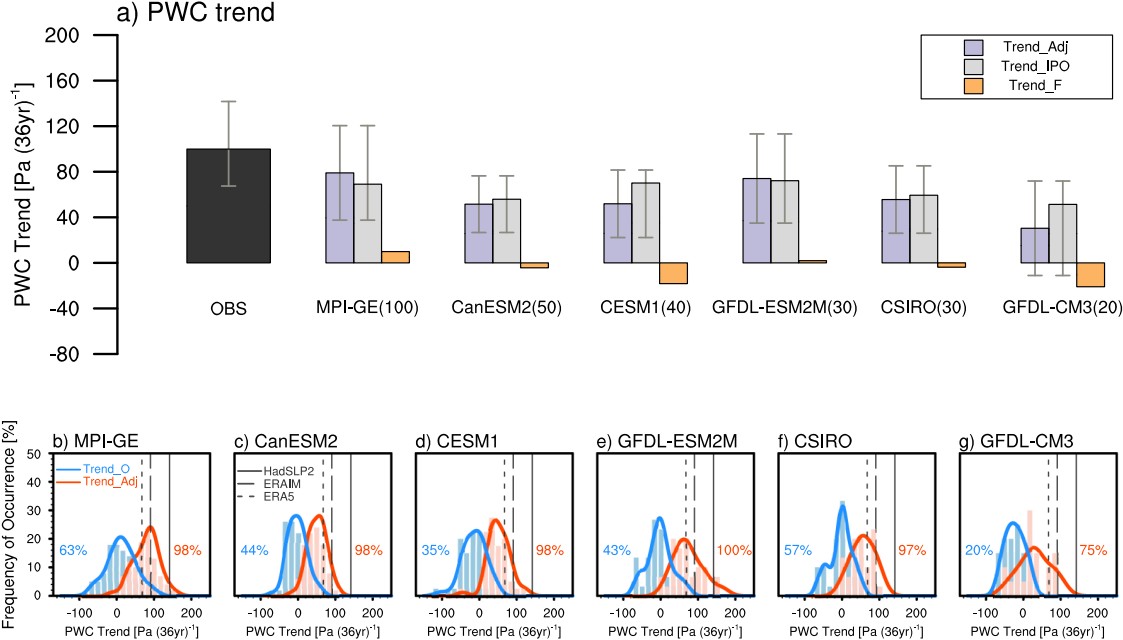

**Fig. 3 Adjustments of PWC trends during 1980–2015 according to the observational IPO phase transition. a** From left to right: the average of observed PWC trend, total PWC trends after adjustment (Trend_Adj, purple bars), IPO-related PWC trends (Trend_IPO, gray bars), forced PWC trends (Trend_F, orange bars) [units: Pa (36 year)$^{-1}$] obtained from MPI-GE, CanESM2, CESM1, GFDL-ESM2M, CSIRO, GFDL-CM3, separately. Numbers in parentheses indicate the ensemble size. Error bars in observation and different large ensembles represent the full observational spread and one standard deviation ensemble member spread, respectively. **b**–**g** Histograms (bars) and fitted distribution (lines) of PWC index trends [units: Pa (36 year)$^{-1}$] derived from different large ensembles. The blue (red) bars and fitted curve show the frequency of occurrence [units: %] of the PWC index trends before (after) the adjustments. Black lines denote the observed PWC index trends derived from HadSLP2, ERAIM, and ERA5. The numbers in percentage indicate the percentage of members showing positive PWC trends with corresponding colors. Differences between the original and adjusted distributions for all the models are significant at 99% confidence level.

After adjusting the individual member-simulated IPO trends to the observed IPO trend during 1980–2015, the IPO phase evolution can contribute about 63% (~51–72%) of the observed PWC strengthening. The dominance of the IPO phase evolution could last into the near-future. The PWC is projected to weaken in the coming decades if we use the IPO phase predicted by the most skillful ensemble members to constrain the PWC projection. Hence, a weakening of the hydrological cycle is expected in the next several decades over the Amazon basin and the Maritime Continent, where the climate anomalies are strongly dominated by PWC changes[41–44]. Consequently, the northern part of the western Amazon is likely to experience a drier climate and less rainfall is expected over large areas of the Maritime Continents. The above results are obtained under the high-end emission scenario of RCP8.5, we extend the analysis to the medium-emission scenario RCP4.5 (Supplementary Fig. 6) and found that the results are not significantly different from that of RCP8.5 (Fig. 4 and Supplementary Fig. 6). This is expected since the near-term climate projections are not very sensitive to plausible alternative scenarios for greenhouse gas concentrations and the externally forced signal of near-term climate change is weaker compared to natural internal variability[45]. Our findings highlight the importance of the IPO-related internal variability to the PWC change and add confidence in an unambiguous attribution of the PWC change. Hence, improving the prediction of IPO and other internal modes of climate variability could reduce the uncertainties in the PWC projection. Reliable predictions and projections of the PWC and the correspondence climate and hydrological cycle change will provide useful information for monitoring programs and guide policy makers for relevant new measures.

In addition, the imbalance between the forced response to aerosol forcing (which tends to strengthen the PWC) and GHG forcing (which tends to weaken the PWC) in climate models may obscure the role of external forcing[12,32]. Although spread exists in the externally-forced response among different models for the past (Fig. 3a), external forcing is likely to amplify the negative PWC trends in the near-future (Fig. 4a and Supplementary Fig. 7). This is because CMIP5 models tend to project an El Niño-like warming under strengthened GHG forcing, with reduced east-west Pacific SST gradient[31,46]. To quantify the contribution of internal variability and external forcing to the near-term changes of PWC, we first calculate the magnitude of total PWC trends shown in Fig. 4a, and also that caused by internal variability. The external forcing is excluded via removing ensemble mean of trends from all large ensemble members. The variability is then calculated by standard deviation of the temporal evolution of the 36-year running trends. We measure the contribution of internal variability to the total change by the ratio between the two estimated magnitudes. We find that internal variability contributes about 71% to the total magnitude of the 36-year running trends, while external forcing makes up the rest, which indicates the dominant role of internal variability. Hence, the IPO phase reversal from negative to positive superimposes on the forced El Niño-like pattern, leading to the weakening of the PWC in the coming decades.

## Methods
**Data and models.** SLP observations for 1950–2015 are taken from the Hadley Centre's globally-complete monthly historical MSLP product (HadSLP2)[47]. To verify the decadal change of the PWC, we also use the time series of SLP derived from the NOAA-CIRES 20th century reanalysis Version 2c (20 CR)[48] and the ECMWF ERA-20C reanalysis[49] over the period 1950–2010. Moreover, we examine the PWC change using reanalysis data including ERA Interim (ERAIM)[50] and ERA5[51] after the satellite era (1980–2015). We exclude SLP data of International Comprehensive Ocean-Atmosphere Data Set (ICOADS) from this study, because

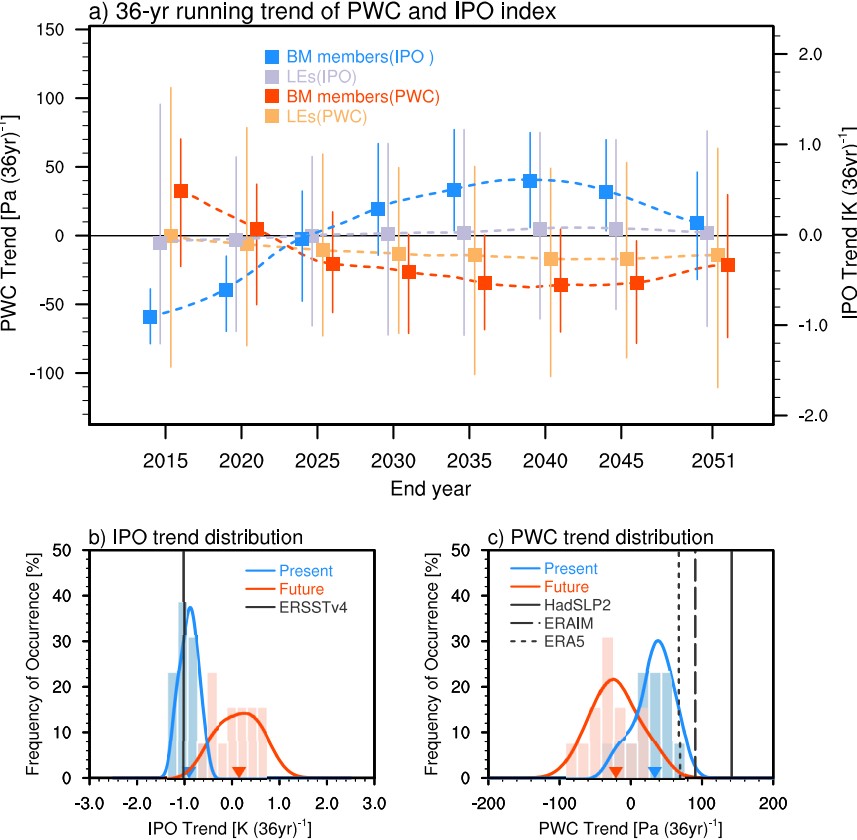

**Fig. 4 Effect of IPO on the near-term projection of the PWC under the RCP8.5 scenario. a** The 36-year running trends in PWC index [units: Pa (36 year)$^{-1}$] and IPO index [units: K (36 year)$^{-1}$] during 1980–2051 obtained from six large ensembles (LEs). Dashed Lines with squares in purple (orange) and blue (red) indicate the ensemble mean IPO (PWC) index trends by LEs and the best match members (BM members), with error bars representing the ensemble member spread. The horizontal axis marks the end year for the 36-year segment. **b** Histograms (bars) and fitted distribution (lines) of IPO index trends [units: K (36 year)$^{-1}$] derived from the BM members. The blue (red) bars and fitted curve show the frequency of occurrence [units: %] of the IPO index trends for present (future) climate. Black line shows the observed IPO index trend derived from ERSST v4. The red and blue triangles denote the ensemble mean of the distribution with the corresponding colors. **c** The same as **b** but for PWC index trends [units: Pa (36 year)$^{-1}$]. Black lines denote the observed PWC index trends derived from HadSLP2, ERAIM, and ERA5. Differences between the future and present distributions for IPO index trends and PWC index trends are significant at 99% confidence level.

previous studies have shown obvious bias of ICOADS SLP data in presenting the Pacific Walker circulation[52]. Monthly wind data are taken from the ERAIM reanalysis data set. Observed monthly SST data are taken from the National Oceanic and Atmospheric Administration/National Climatic Data Center Extended Reconstructed SST version 4 (ERSST v4), covering the period 1901-2017 with a horizontal resolution of 2.0° × 2.0°[53].

To determine the relative roles of external forcing and internal variability, we used the outputs from the Max Planck institute Grand Ensemble (MPI-GE), with 100 members for the historical simulation[54]. All MPI-GE simulations share a single model MPI-ESM1.1 (version MPI-ESM 1.1.00p2), which is run in low-resolution configuration of an atmosphere of approximately 1.9° and an ocean of approximately 1.5° resolution. All MPI-GE members have the same external forcing following the phase 5 of the Coupled Model Intercomparison Project (CMIP5)[55] protocol as well. Since individual ensemble members only differ in their atmospheric initial conditions, internal variability is the only source of uncertainty in MPI-GE simulations. More information about the MPI-GE design is provided in ref. [37]. Although previous studies suggest that current state-of-the-art coupled climate models failed to reproduce the observed PWC strengthening trend[10,12,15], which might be a result of internal variability, MPI-GE performs well in capturing the recent PWC change (Supplementary Fig. 8).

We also use the outputs of another five large ensembles form a new collection of initial-condition large ensembles generated with different Earth system models (ESM-LE) under historical and future radiative forcing scenarios, including CanESM2, CESM1, GFDL-ESM2M, CSIRO-MK3.6, and GFDL-CM3. All models are capable to simulate the IPO-PWC relationship (Supplementary Figs. 8 and 9). More details about ESM-LE model outputs collected from the US CLIVAR Working Group on Large Ensembles can be found in http://www.cesm.ucar.edu/projects/community-projects/MMLEA/.

A common period 1980–2015 for these six large ensembles are selected, in consistent with the observations. The overall period 1980–2015 is connected by the period 1980–2005 from the historical simulations and the period 2006–2015 from

the RCP8.5 simulations. The period 2016–2051 under the RCP8.5 scenario are used for projection. A 1021-year data from the pre-industrial control simulation of MPI-GE is used to verify the influence of the internal variability. In addition, the period 1980–2051 under the RCP4.5 scenario from MPI-GE is also used.

**Separating the externally-forced signal and internal variability.** A 9-year running mean was applied to raw data in the observations and models to extract the interdecadal signal firstly. Since different members from the same ensemble simulation are driven by the same external forcing, the ensemble mean of all members can be regarded as the response to external forcing and deviations in each ensemble member from the ensemble mean can be taken as arising from the internal variability. Thus, for a certain variable X, the separation in member i can be expressed as:

$$X(i) = X_{forced} + X_{internal}(i), i = 1, 2, \ldots, n-1, n \qquad (1)$$

where $X_{forced}$ is the ensemble mean of different members, denoting the response to external forcing. $X_{internal}(i)$ is the residual of the original $X(i)$ minus the forced response $X_{forced}$, which varies among different members and shows the variability associated with internal variability.

**Definition of the IPO, AMO and PWC index.** In this study, we define the IPO index as the 9-year running mean of the difference in averaged annual mean SST anomalies between the tropical central-eastern Pacific (80°E to 90°W, 10°S to 15°N) and the north Pacific (150°E to 160°W, 30°N to 45°N) following ref. [36]. The IPO index for each ensemble member is calculated by the internal part of SST, that is $SST_{internal}(i)$ (see Methods, Eq. (1)). Specifically, the time series of the IPO index for member i [$IPO(i, t)$], are calculated as the 9-year running mean of north-south gradients of the area-averaged annual mean $SST_{internal}(i)$ anomalies. The IPO-related SST anomalies are obtained by regressing the $SST(i)$ with respect to the

$IPO(i, t)$. We define the AMO index as the 9-year running mean of the area-averaged annual mean SST aomalies over the North Atlantic Ocean (80°W to 0°E, 0°N to 65°N) after subtracting the global mean (180°W to 180°E, 80°S to 80°N) SST anomaly time seires following ref. [36].

To estimate the strength of the PWC, we define the PWC index as the east–west SLP gradient along the equatorial Pacific based on the difference of the area-averaged SLP between 5°S–5°N, 160°W–80°W, and 5°S–5°N, 80°E–160°E[12,35]. Positive values represent an enhanced Indo-Pacific SLP gradient, indicating a strengthened PWC.

**Adjustment of PWC trends based on IPO.** Following ref. [56], the contribution of IPO to the PWC trend of member $i$ for the period $\tau$ (here $\tau = 1980$–$2015$) on the multi-decadal time scale can be expressed as

$$\frac{\partial PWC_{IPO}(i)}{\partial t} = r_{PWC,IPO}(i) \bullet \frac{\partial IPO(i)}{\partial t}, i = 1, 2, \ldots, n - 1, n \quad (2)$$

$$r_{PWC,IPO}(i) = \frac{\partial PWC(i)}{\partial IPO(i)}, \quad (3)$$

where $r_{PWC,IPO}(i)$ is the regression coefficient of the 9-year running-mean PWC index regressed onto the IPO time series of member $i$ over the period of 1980–2015. $\frac{\partial IPO(i)}{\partial t}$ is the trend of the IPO index of member $i$, representing the phase transition of IPO during the analysis period $\tau$. $\frac{\partial PWC_{IPO}(i)}{\partial t}$ is the IPO-related PWC trend of member $i$ over the same period and varies among the ensemble members.

To quantitatively estimate the effect of IPO phase transition on the PWC change, we first adjust the model-simulated IPO phase among various members to the observation. After the adjustment, all the ensemble members can be viewed as being regulated by the same observational IPO phase evolution rather than the random IPO phase transitions. By adding an adjustment term to the original PWC trend following refs. [39,56], the adjusted linear trend of the PWC is given by

$$\frac{\partial PWC_{adj}(i)}{\partial t} = \frac{\partial PWC_{forced}}{\partial t} + \frac{\partial PWC_{internal\_adj}(i)}{\partial t}, i = 1, 2, \ldots n - 1, n, \quad (4)$$

where the adjusted PWC trend of member $i$ [$\frac{\partial PWC_{adj}(i)}{\partial t}$] is the sum of the forced PWC trend [$\frac{\partial PWC_{forced}}{\partial t}$] plus the internal component of the PWC trend [$\frac{\partial PWC_{internal}(i)}{\partial t}$] with an adjustment term [$\frac{\partial PWC_{internal\_adj}(i)}{\partial t}$]. The internal adjusted PWC trend is expressed as

$$\frac{\partial PWC_{internal\_adj}(i)}{\partial t} = \frac{\partial PWC_{internal}(i)}{\partial t} + \alpha_{internal}(i), \quad (5)$$

where

$$\alpha_{internal}(i) = -r_{PWC,IPO}(i) \bullet \left( \frac{\partial IPO(i)}{\partial t} - \frac{\partial IPO_{OBS}}{\partial t} \right), \quad (6)$$

where $\alpha_{internal}(i)$ is the adjustment term and takes the observed IPO phase transitions into consideration. Hence Eq. (4) can be written as

$$\frac{\partial PWC_{adj}(i)}{\partial t} = \frac{\partial PWC_{forced}}{\partial t} + \frac{\partial PWC_{internal}(i)}{\partial t} - r_{PWC,IPO}(i) \bullet \left( \frac{\partial IPO(i)}{\partial t} - \frac{\partial IPO_{OBS}}{\partial t} \right), i = 1, 2, \ldots n - 1, n, \quad (7)$$

Based on Eq. (7), the ensemble mean of $\frac{\partial PWC_{adj}(i)}{\partial t}$ represents the PWC change in response to external forcing and the observed IPO phase transition. The contribution of the observed IPO phase transition is estimated by the relative percentage of the IPO-related PWC trend to the observed PWC trend.

**Significant test.** In this study, the significance of trends was tested by the Mann–Kendall nonparametric method. The Monte Carlo nonparametric method is used to test the significance of the regression coefficients onto the filtered time series. The Kolmogorov–Smirnov nonparametric test is applied to determine whether two probability distributions are well-distinguished.

## Data availability

The six LEs analyzed in this study are available from the Multi-Model Large Ensemble Archive [https://www.cesm.ucar.edu/projects/community-projects/MMLEA/]. Observational SLP data HadSLP2 is from Met Office Hadley Centre [https://www.metoffice.gov.uk/hadobs/hadslp2/]. ERSST v4 and 20CR are provided by NOAA Physical Sciences Laboratory [https://psl.noaa.gov/data/gridded/index.html]. ERA-20C, ERAIM, and ERA5 are available from ECMWF [https://www.ecmwf.int/en/forecasts/datasets/browse-reanalysis-datasets].

## Code availability

The data in this study is analyzed with NCAR Command Language (NCL; http://www.ncl.ucar.edu/). All relevant codes used in this work are available, upon request, from the corresponding author T.Z.

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

## Acknowledgements

The study is supported by National Natural Science Foundation of China under Grant Nos. 41988101, 41775091, and K. C. Wong Education Foundation. C. L. was supported by the Clusters of Excellence CLICCS (EXC2037), University of Hamburg, funded by the German Research Foundation (DFG). X. C. was supported by the National Key Research and Development Program of China (2020YFA0608904). We acknowledge the support from Jiangsu Collaborative Innovation Center for Climate Change. We thank Dr. Clara Deser from the National Center for Atmospheric Research in Boulder USA for her very helpful comments and suggestions on observational data quality control. We thank the US CLIVAR Working Group on Large Ensembles for collecting and making available the multi-model large ensemble archive (MMLEA, http://www.cesm.ucar.edu/projects/communityprojects/MMLEA/).

## Author contributions

T.Z. designed the research, provided comments, and revised the manuscript. M.W. performed the analysis and drafted the manuscript. C.L. and H.L. helped organize and revise the draft. X.C., B.W., W.Z., and L.Z. gave comments and contributed to the discussion of the results. All of the co-authors contributed to scientific interpretations and subsequent revisions.

## Competing interests

The authors declare no competing interests.
