## [Peer Review File · Nature Communications]

REVIEWER COMMENTS

Reviewer #1:

Review of "A very likely weakening of Pacific Walker Circulation in constrained near-future projections" by Wu et al.

The central goal of this paper is to use the observed (projected) IPO phase transitions to constrain climate model simulations (near-term projections), in order to estimate the effects of the IPO and external forcing on the Pacific Walker Circulation (PWC) changes. The paper includes two parts: Part I assesses the relative roles of the IPO and external forcing in causing the intensified PWC from 1980-2015 (36yrs), and the authors conclude that the IPO accounts for 68% of the observed PWC strengthening; Part II uses 13 BM members (that match the observed IPO phase evolution with significant IPO-PWC correlations across 6 LE models) to project the IPO phase in the next 36yrs from 2016-2051, and they conclude that the IPO will transit to its positive phase and the PWC is very likely to weaken.

The approach of constraining the IPO phase transitions using observations and BM-member projections is scientifically interesting, and investigating ways to achieve more reliable near-term projections of PWC change is climatically important. The results presented here - particularly the near-term projection part - however do not seem convincing at least at the present stage, because of the weak IPO changing signals and the large uncertainty of external forcing across different models. My comments are detailed below, and hope they are helpful for improving the MS.

Comments

Part I: 1985-2015. There are quite a few studies on the role of the IPO in controlling the PWC using observational analysis and modelling experiments, and some studies cover most of the 1980-2015 period examined in this MS and the recent global surface warming hiatus period (e.g., Meehl et al. 2011; Sohn et al. 2013; England et al. 2014; Han et al. 2017; Dong et al. 2018). These relevant studies will not devalue this MS, because the authors here use a "constraint" approach which might have not been applied by any of the existing studies.

Meehl et al., 2011: Model-based evidence of deep-ocean heat uptake during surface-temperature hiatus periods. *Nature Climate Change*.

Sohn et al. 2013: Observational evidence of Walker circulation changeover the last 30 years contrasting with GCM results. *Clim Dyn*.

Han et al. 2017: Decadal variability of Indian and Pacific Walker Cells: Do they co-vary on decadal timescales? *J. Clim*.

Dong et al. 2018: Asymmetric Modulation of ENSO Teleconnections by the Interdecadal Pacific Oscillation. *J. Clim*.

Part II: Near-term projection for 2016-2051.

(1) The BM members that match the observed IPO phase for the 1950-1985 period indeed capture the observed negative IPO transition from 1986-2015 (Supporting Fig

5a), but they failed to simulate the PWC intensification for the same period (Supporting Fig 5b), which is attributed primarily to the IPO. This makes me wonder whether the 13 BM members chosen based on 1980-2015 can realistically capture the projected PWC in the upcoming decades.

- (2) The 6 LE climate model experiments show large spread in simulating the PWC change induced by external forcing during 1980-2015, from weakened to intensified PWC (Fig 3a). While climate models tend to produce weakened PWC under GHG forcing (e.g., Vecchi & Soden 2007), observations show evidence of enhanced PWC on multi-decadal & centennial timescales (e.g., Solomon and Newman 2012; L'Heureux et al. 2013; Newman 2013), due to the thermostat mechanism (Clement et al. 1996; Zhang et al. 2019).

Note that the MPI-GE (100members) is the best model that realistically simulates the enhanced PWC from 1980-2015 (Fig 3a), followed by GFDL-ESM2M (30members). The authors may want to separately examine the PWC's response for both the present and future periods using the MPI-GE + GFDL-ESM2M (total of 130members) & compare them against the rest models (140members). Averaging all 270members essentially eliminated the enhanced PWC response from the MPI-GE, which may be physically correct, and produced a weakened PWC. Separately assessing each model will also be helpful, even though some models may not have members with IPO phases matching the observed.

- (3) The weakened PWC in the 2016-2051 projections seems result largely from external forcing (Fig 4a, compare the grey and red). After removing the grey, seems that the weakened PWC due to IPO is rather weak and may not be statistically significant. Fig 4b shows that the IPO index is changing from negative to positive after the 2015-16 El Nino; however it remains close to 0 after 2020. Is the IPO trend from 2020-2051 (red curve of 4b) statistically significant? How much the weakened PWC is a response to GHG forcing (Fig 4d)? Seems that external forcing is included in 4d. Again, separately examining the MPI-GE & GFDL-ESM2M against others might be beneficial.

Solomon, A., and M. Newman, 2012: Reconciling disparate twentieth-century Indo-Pacific ocean temperature trends in the instrumental record. *Nat. Climate Change*

Newman, M., 2013: Atmospheric science: Winds of change. *Nat. Climate Change*

L'Heureux, M., S. Lee, and B. Lyon, 2013: Recent multi-decadal strengthening of the Walker circulation across the tropical Pacific. *Nat. Climate Change*

Zhang et al. 2019: Indian Ocean Warming Trend Reduces Pacific Warming Response to Anthropogenic Greenhouse Gases: An Interbasin Thermostat Mechanism. *Geophys. Res. Lett.*

- (0) Lines 125-128, reference 19:

Note that Wang et al. 2013 demonstrated that the 3yr running mean AMO and IPO indices are uncorrelated with $r=0.07$ from 1958-2011, but both impacts NHSM. Many

existing studies however do show AMO impacts on PWC (e.g., Kucharski et al. 2015; Johnson et al. 2020; Kim et al. 2020). Are they examining interannual timescale without filtering, whereas this MS examines "decadal scale" variability with 9yr running mean?

Kucharski et al, 2015: Atlantic forcing of Pacific decadal variability. *Clim Dyn.*

Johnson et al. 2020: Pacific decadal oscillation remotely forced by the equatorial Pacific and the Atlantic Oceans. *Clim Dyn.*

Kim et al. 2020: Pacific Mean-State Control of Atlantic Multidecadal Oscillation–El Niño Relationship. *J. Clim.*

Reviewer #2 (Remarks to the Author):

Review's comments for the paper (NCOMMS-21-10064-T), entitled "A very likely weakening of Pacific Walker Circulation in constrained near-future projections", submitted to *Nature Communications*

By using both observations and large ensemble simulations from multimodel simulations, this paper investigated the recent enhancement of the Pacific Walker circulation (PWC) and made constrained near-term projections. The analyses suggested that recent enhanced PWC is very likely the internal variability associated with a negative trend of the interdecadal Pacific Oscillation (IPO). The constrained projections by IPO phase change suggested a high probability of a weakened PWC in the near future. Results are interesting and the paper is worth of publishing. However, the paper needs some improvements by properly addressing the comments listed below before it can be accepted for publication in *Nature Communications*.

Major comments

1. One major concern the reviewer has is the assurance of the study that the interdecadal Pacific Oscillation (IPO) is driving the interdecadal change of Pacific Walker circulation (PWC). The analyses of both observations and model simulations for historical simulations suggest an association of trends between IPO and PWC with strengthened PWC being accompanied with negative trends of IPO. However, there is no evidence to support the authors' assurance that it is IPO change driving PWC. Authors need to give more convincing evidence to support this assurance.
2. Another important feature from various analyses is the association between changes in PWC and zonal sea surface temperature (SST) gradient in the tropical Pacific with recent strengthened PWC being associated with enhanced zonal SST gradient. There is a wealth of literature about PWC variability and zonal SST gradient variability on interannual to interdecadal time scales. In the paper, it mentioned the role of zonal SST gradient in the tropical Pacific. A question that arises is therefore how IPO, zonal SST gradient, and PWC changes (trends) are related to each other. Is the IPO affecting PWC through its related zonal SST gradient in the tropical Pacific? If it is, this should be made more clearly in the paper.

Minor comments

1. Lines 71-72. See major comment 1.

2. Lines 78-79. Quantify the forced trend.
3. Lines 97-122. It is not clear to the reviewer how the authors get this conclusion that IPO regulates PWC variability since all these analyses only demonstrate an association between two phenomena and do not suggest the causality. See major comment 1.
4. Lines 147-148. What is the relationship between zonal SST gradient and IPO? See major comment 2.
5. Lines 175-178 and Lines 184-187. They give similar information.
6. Lines 179-180. If the reviewer understood correctly, Frequency of occurrence (FOC) shown in figures 4c and d are based on 13 members (BM). Are these FOCs robust since you used such small ensemble members to construct them?
7. Lines 198-199. See major comment 1.
8. Lines 244-245. "MPI-GE shows reasonable performance in capturing the recent PWC change". This statement is confusing. Rephrase.
9. Line 289. "in for"?
10. Figures. In some figures. Probability density function(PDF) is used while frequency of occurrence (FOC) is used in others. It seems to the reviewer that they are essentially the same thing and would be better to be consistent.

Reviewer #3 (Remarks to the Author):

The manuscript "A very likely weakening of Pacific Walker Circulation in constrained near-future projections" attempts to make a projection of the Walker Circulation over roughly the next 30 years. The study uses coupled models, which are known to underestimate the observed recent strengthening of the Walker Circulation. To remedy this underestimation, they statistically adjust the models to the observed trend to assess the relative contributions of internal vs. external forcings. Furthermore, they use the models that vary closely with the observations for near future projections. The topic is interesting and worth pursuing, however, some aspects of this study require further development and clarification.

A large limiting factor of the study is the notion that the models can be adjusted or chosen to better represent the observations, however, this does not address the very real possibility that the models are not representing the physics correctly on these timescales (see citations 10, 11, 12 in manuscript). I do not expect this study to remedy the physics in the models, but some of the issues I have with this study are the result of these model deficiencies.

There are two points that give me reservations in recommending publication of this paper, and a number of subpoints for each. However, I would like to give the authors an attempt at addressing these issues for a possible future submission.

1. The method underpinning their result of the likely weakening of the Pacific Walker Circulation does not perform well in their verification analysis. In Line 173 the authors state "[t]he weakening of the PWC from the 1950s to 1980s and the strengthening thereafter are well reproduced", however Sup. Figs. 5b and 5d do not show a strengthening that is comparable to that observed. The red line in Sup. Fig. 5b does not follow the observed black line, and the observed trend in Sup. Fig. 5d is well outside any modelled trend during 1986-2015. This gives doubt to the results shown in Fig. 4a and Fig.4d, which underpins the statement given in the manuscript title.

- 1.1. What is the difference in choosing members that are in the same phase, compared to members that have a similar IPO value at the end of your calibration period? I don't

think we know enough about the IPO that we can rely on its current oscillation to continue.

1.2. Line 166: State in the main text how you choose 13 members. Also, which models do these members come from?

2. The adjustment used to make the conclusion that 68% of the observed PWC strengthening is due to the IPO needs to be clarified. The adjustment relies on the relationship between 9-year running mean PWC and IPO from 1980-2015, but this relationship is not shown for the models or observations. Are the models realistic in this relationship? Likewise, how do the models perform in simulating the IPO trends compared to observed? My reservations are that if the models cannot simulate an PWC-IPO relationship or IPO trend of similar magnitude to the observations, can the adjustment made to the models be believed?

2.1. In Figure 3a, why are the forced trends different between models? Considering the models have the same forcing, I would assume you would get a consistent response. I don't follow why it is due to the model spread of the equatorial zonal SST gradient mentioned in line 147 and the figures you reference here.

2.2. Can you clarify how $\partial \text{PWC}_{\text{internal}(i)} / \partial t$ is calculated in Equation 7?

2.3. Line 149: The adjusted PWC trend and observed PWC trend do not seem "close" to me, their means (or medians?) are not in the other's range. A statistical test would help this disagreement.

2.4. Fig. 2 e and f: To help address my second main point, you could include an observational point, and provide these scatterplots for other models.

2.5. Supp. Figure 2e: Provide this plot for the IPO, it would also help address my second main point. If the running correlation of PWC and IPO shows the same variation as shown in this figure, then it would have implications on the adjustment. Along these lines, if there is a relationship between the IPO-PWC relationship and the IPO trend, could you make a more accurate adjustment?

Response to reviewers' comments of NCOMMS-21-10064-T "A very likely weakening of Pacific Walker Circulation in constrained near-future projections"

We wish to express our appreciation to the reviewers for all the constructive comments that helped us to improve the manuscript. We have addressed the comments from the three reviewers and revised the manuscript based on their suggestions. Please see our point-by-point response below. In the following, the reviewer's comments are written in black, followed by our response in blue.

Response to Reviewer #1:

The central goal of this paper is to use the observed (projected) IPO phase transitions to constrain climate model simulations (near-term projections), in order to estimate the effects of the IPO and external forcing on the Pacific Walker Circulation (PWC) changes. The paper includes two parts: Part I assesses the relative roles of the IPO and external forcing in causing the intensified PWC from 1980-2015 (36yrs), and the authors conclude that the IPO accounts for 68% of the observed PWC strengthening; Part II uses 13 BM members (that match the observed IPO phase evolution with significant IPO-PWC correlations across 6 LE models) to project the IPO phase in the next 36yrs from 2016-2051, and they conclude that the IPO will transit to its positive phase and the PWC is very likely to weaken.

The approach of constraining the IPO phase transitions using observations and BM member projections is scientifically interesting, and investigating ways to achieve more reliable near-term projections of PWC change is climatically important. The results presented here - particularly the near-term projection part - however do not seem convincing at least at the present stage, because of the weak IPO changing signals and the large uncertainty of external forcing across different models. My comments are detailed below, and hope they are helpful for improving the MS.

We would like to express our appreciation to the reviewer for all the constructive comments that helped us to improve the manuscript. According to the reviewer's suggestions, we have thoroughly revised the manuscript. We believe that results from the 36-year running trend of the IPO and PWC index and assessing models separately help to further convince readers of our conclusions.

Comments

Part I: 1985-2015. There are quite a few studies on the role of the IPO in controlling the PWC using observational analysis and modelling experiments, and some studies cover most of the 1980-2015 period examined in this MS and the recent global surface warming hiatus period (e.g., Meehl et al. 2011; Sohn et al. 2013; England et al. 2014; Han et al. 2017; Dong et al. 2018). These relevant studies will not devalue this MS, because the authors here use a "constraint" approach which might have not been applied by any of the existing studies.

Meehl et al., 2011: Model-based evidence of deep-ocean heat uptake during surface-temperature hiatus periods. *Nature Climate Change*.

Sohn et al. 2013: Observational evidence of Walker circulation changeover the last 30 years contrasting with GCM results. *Clim Dyn*.

Han et al. 2017: Decadal variability of Indian and Pacific Walker Cells: Do they co-vary on decadal timescales? *J. Clim*.

Dong et al. 2018: Asymmetric Modulation of ENSO Teleconnections by the Interdecadal Pacific Oscillation. *J. Clim*.

Thank you very much for these encouraging comments. The recommended references have been cited.

Part II: Near-term projection for 2016-2051.

(1) The BM members that match the observed IPO phase for the 1950-1985 period indeed capture the observed negative IPO transition from 1986-2015 (Supporting Fig 5a), but they failed to simulate the PWC intensification for the same period (Supporting Fig 5b), which is attributed primarily to the IPO. This makes me wonder whether the 13 BM members chosen based on 1980-2015 can realistically capture the projected PWC in the upcoming decades.

Thank you for your insightful comment. In the previous version of the submission, we showed the annual mean time series of the IPO index and PWC index (Fig. 4 and Supplementary Fig. S5 in the previous manuscript). To highlight the decadal variation, in the revised manuscript, we calculate the 36-year running trend of the IPO index and PWC index (Fig.4 and Fig.S5 in the revised manuscript). To verify the method of selecting members, we choose BM members based on 1950-1985 to see whether they can reproduce the observed PWC change in 1980-2015. The result shows that although the magnitude of the PWC change simulated by the selected BM members is weaker than the observation, the observed strengthening of the PWC with increasing trends are indeed captured (Fig.S5a), which adds creditability of this method.

The error bars shown in Fig.S5a represent one standard deviation of the spread among the ensemble members and the full uncertainty range is indicated by shading. The observed PWC trend generally lays within the range of the ensemble member spread, indicating models' ability to reproduce the observed PWC variation (Supplementary Fig. 5a). In addition, among the selected 13 members we do find some individual realizations match the observational tendency of PWC better than the ensemble mean shown here. This highlights the importance of even larger ensemble size, which would allow us to choose more BM members.

Based on your comments, we have modified the relevant figures (Fig. 4 and Supplementary Figs. 4-5) and text (L158, L160, L170-171, L176-179).

L170-171: “To access the credibility of this method, we choose BM members based on 1950-1985 period to see whether they can reproduce the observed PWC change in 1980-2015”.

L176-179: “Noted that although the selected BM members simulate a smaller magnitude of the PWC change compared with that in the observation, the observed PWC trend generally lays within the range of the ensemble member spread, indicating models’ ability to reproduce the observed PWC variation (Supplementary Figs. 5a).”

Supplementary Figure 5 | Verifying the method of selecting ensemble members according to observed IPO. a. The 36-year running trends in PWC index [units: $\text{Pa} (36 \text{ year})^{-1}$] and IPO index [units: $\text{K} (36 \text{ year})^{-1}$] during 1950-2015 obtained from six LEs. Dashed Lines with squares in purple (orange) and blue (red) indicate the ensemble mean IPO (PWC) trends by LEs and the best match members, with error bars representing one standard deviation ensemble member spread. Black and gray lines are from observations. Purple (orange) shading denotes the range across all the ensemble members for IPO (PWC) trends. The horizontal axis marks the end year for the 36-year segment. **b.** Histograms (bars) and fitted distribution (lines) of IPO index trends [units: $\text{K} (36 \text{ year})^{-1}$] derived from the best match members. The blue (red) bars and fitted

curve show the frequency of occurrence [units: %] of the IPO index trends for the 1950-1985 (1980-2015) period. The blue and red dashed lines show the observed IPO index trend for the two periods. The blue and red triangles denote the ensemble mean of the distribution with the corresponding color. **c.** The same as **b.** but for PWC index trends [units: Pa (36 year)⁻¹]. Differences between the distributions for the 1980-2015 and 1950-1985 periods are significant at 99% confidence level.

(2) The 6 LE climate model experiments show large spread in simulating the PWC change induced by external forcing during 1980-2015, from weakened to intensified PWC (Fig 3a). While climate models tend to produce weakened PWC under GHG forcing (e.g., Vecchi & Soden 2007), observations show evidence of enhanced PWC on multi-decadal & centennial timescales (e.g., Solomon and Newman 2012; L'Heureux et al. 2013; Newman 2013), due to the thermostat mechanism (Clement et al. 1996; Zhang et al. 2019). Note that the MPI-GE (100members) is the best model that realistically simulates the enhanced PWC from 1980-2015 (Fig 3a), followed by GFDL-ESM2M (30members). The authors may want to separately examine the PWC's response for both the present and future periods using the MPI-GE + GFDL-ESM2M (total of 130members) & compare them against the rest models (140members). Averaging all 270 members essentially eliminated the enhanced PWC response from the MPI-GE, which may be physically correct, and produced a weakened PWC. Separately assessing each model will also be helpful, even though some models may not have members with IPO phases matching the observed.

Thank you for the excellent suggestion. Although climate models tend to project weakened PWC under GHG forcing on the long-term timescale (Vecchi et al. 2006), the role of external forcing in the recent strengthening of the PWC remains inconclusive. This is partly because the balance between the forced response to aerosol forcing which tends to strengthen the PWC, and GHG forcing which tends to weaken the PWC may not be correct in climate models (DiNezio et al. 2013; Kociuba and Power 2015). Models may overestimate the weakening of the PWC caused by GHG forcing or underestimate the strengthening of the PWC caused by aerosol forcing, leading to large model spread in the recent PWC change in response to external forcing.

As suggested, we first divide six LEs into two groups: Group-S (MPI-GE and GFDL-ESM2M, 130 members in total), in which external forcing induced a strengthened PWC during 1980-2015, and Group-W (CanESM2, CESM, CSIRO and GFDL-CM3, 140 members in total), in which external forcing induced a weakened PWC. We then compare the simulated and projected PWC change for both the present (1980-2015) and future (2016-2051) periods from those two groups (see Supplementary Fig. 6 below). The results show that in both LE groups, the BM members can reproduce the IPO phase shift from positive to negative along with the strengthening of the PWC at present (Supplementary Figs. 6a and c). In the near-future, both LE groups project a negative-to-positive IPO phase transition, as well as the weakening of the PWC (Supplementary Figs. 6b and d). As for external forcing, it plays a positive role in the PWC strengthening during the historical period in Group-S, while it tends to weaken

the PWC in Group-W (Supplementary Figures 6a and c). In the near-future, however, external forcing is likely to amplify the negative PWC trends in both LE groups (Supplementary Figures 6b and d). The above results confirm the robustness of the method of selecting BM members and the conclusion that the PWC is very likely to weaken, which is mainly related to the recovery of IPO in the coming decades.

Supplementary Figure 6 | Role of external forcing. The 36-year running trends in PWC index [units: $\text{Pa} (36 \text{ year})^{-1}$] and IPO index [units: $\text{K} (36 \text{ year})^{-1}$] during (left) 1950-2015 and (right) 1980-2051 obtained from **a-b.** MPI-GE and GFDL-ESM2M and **c-d.** CanESM2, CESM, CSIRO and GFDL-CM3. Dashed Lines with squares in purple (orange) and blue (red) indicate the ensemble mean IPO (PWC) index trends by LEs and the best match members, with error bars representing one standard deviation ensemble member spread. The horizontal axis marks the end year for the 36-year segment.

References

Vecchi, G. A., B. J. Soden, A. T. Wittenberg, I. M. Held, A. Leetmaa, and M. J. Harrison, 2006: Weakening of tropical Pacific atmospheric circulation due to anthropogenic forcing. *Nature*, **441**, 73-76.

DiNezio, P. N., G. A. Vecchi, and A. C. Clement, 2013: Detectability of changes in the Walker circulation in response to global warming. *J. Clim.*, **26**, 4038–4048.

Kociuba, G., and S. B. Power, 2015: Inability of CMIP5 Models to Simulate Recent Strengthening of the Walker Circulation: Implications for Projections. *Journal of Climate*, **28**, 20-35.

(3) The weakened PWC in the 2016-2051 projections seems result largely from external

forcing (Fig 4a, compare the grey and red). After removing the grey, seems that the weakened PWC due to IPO is rather weak and may not be statistically significant. Fig 4b shows that the IPO index is changing from negative to positive after the 2015-16 El Nino; however, it remains close to 0 after 2020. Is the IPO trend from 2020-2051 (red curve of 4b) statistically significant? How much the weakened PWC is a response to GHG forcing (Fig 4d)? Seems that external forcing is included in 4d. Again, separately examining the MPI-GE & GFDL-ESM2M against others might be beneficial.

Solomon, A., and M. Newman, 2012: Reconciling disparate twentieth-century Indo-Pacific Ocean temperature trends in the instrumental record. *Nat. Climate Change*.

Newman, M., 2013: Atmospheric science: Winds of change. *Nat. Climate Change*.

L'Heureux, M., S. Lee, and B. Lyon, 2013: Recent mul-tidecadal strengthening of the Walker circulation across the tropical Pacific. *Nat. Climate Change*.

Zhang et al. 2019: Indian Ocean Warming Trend Reduces Pacific Warming Response to Anthropogenic Greenhouse Gases: An Interbasin Thermostat Mechanism. *Geophys. Res. Lett.*

Thanks for this valuable comment. While we agree that external forcing can also affect the PWC change, we hope to highlight that it is internal variability that plays the major role in modulating the present and near-future PWC change. As seen from the revised Fig. 4, the projected PWC change is well correspond to the IPO phase transition (Fig. 4a). The IPO trends calculated based on the 36-year running window change sign from negative to positive during 2016-2051, with reversed sign change of the PWC trends throughout the corresponding period. Similar results are obtained by separately examining models as you suggested (Supplementary Figure 6b and d). To further address this point, we examine the internal IPO and PWC trends by removing the influence of external forcing (i.e. remove each LE's ensemble mean) for all six LEs, Group-S and Group-W (Figure R1 shown below). The results are consistent with that from Fig. 4 and Supplementary Fig. 6, confirming the robustness of our conclusions.

To quantify the contribution of internal variability and external forcing to the near-term changes of PWC, we first calculate the magnitude of total PWC trends shown in Fig.4a, and also that caused by internal variability shown in Fig. R1a. The external forcing is excluded via removing ensemble mean of trends from all large ensemble members. The variability is then calculated by standard deviation of the temporal evolution of the 36-year running trends. We measure the contribution of internal variability to the total change by the ratio between the two estimated magnitudes. We find that internal variability contributes about 71% to the total magnitude of the 36-year running trends, while external forcing makes up the rest. The corresponding contribution for Group-S (Group-W) is 64% (68%). The estimates indicate the dominant role of internal variability.

Based on your comment 2 and comment 3, we have added Supplementary Figure 6 and

discussed the role of external forcing in Lines 217-235 in the main text and Lines 84-101 in the Supplementary information.

L217-235: “In addition, the imbalance between the forced response to aerosol forcing (which tends to strengthen the PWC) and GHG forcing (which tends to weaken the PWC) in climate models may obscure the role of external forcing^{12,32}. Although spread exists in the externally-forced response among different models for the past (Fig. 3a), external forcing is likely to amplify the negative PWC trends in the near-future (Fig. 4a and Supplementary Fig. 6). This is because CMIP5 models tend to project an El Niño-like warming under strengthened GHG forcing, with reduced east-west Pacific SST gradient^{31,45}. To quantify the contribution of internal variability and external forcing to the near-term changes of PWC, we first calculate the magnitude of total PWC trends shown in Fig.4a, and also that caused by internal variability. The external forcing is excluded via removing ensemble mean of trends from all large ensemble members. The variability is then calculated by standard deviation of the temporal evolution of the 36-year running trends. We measure the contribution of internal variability to the total change by the ratio between the two estimated magnitudes. We find that internal variability contributes about 71% to the total magnitude of the 36-year running trends, while external forcing makes up the rest, which indicates the dominant role of internal variability. Hence, the IPO phase reversal from negative to positive superimposes on the forced El Niño-like pattern, leading to the weakening of the PWC in the coming decades”.

Supplementary information, L84-101: “Climate models tend to project weakened PWC under GHG forcing on the long-term timescale¹ but the role of external forcing in recent strengthening of the PWC remains controversial. This is because the balance between the forced response to aerosol forcing which tends to strengthen the PWC, and GHG forcing which tends to weaken the PWC may not be correct in climate models^{2,3}. To explore the role of external forcing, we divide six LEs into two groups: Group-S (MPI-GE and GFDL-ESM2M, 130 members in total), in which external forcing induced a strengthened PWC during 1980-2015, and Group-W (CanESM2, CESM, CSIRO and GFDL-CM3, 140 members in total), in which external forcing induced a weakened PWC. We then compare the simulated and projected PWC change for both the 1950-2015 and 1980-2051 periods from those two groups (Supplementary Fig. 6). The result shows that external forcing is likely to strengthen the PWC in Group-S over the past several decades, while it tends to weaken the PWC in Group-W (Supplementary Figures 6a and c). In the near-future, however, external forcing is likely to amplify the negative PWC trends in both LE groups (Supplementary Figures 6b and d). The internal variability contributes roughly 64% in Group-S and 68% in Group-W to the total magnitude of the averaged 36-year running trends in the projection. Consequently, internal variability superimposes on the forced response and dominates the PWC change”.

Fig. 4. Effect of IPO on the near-term projection of the PWC under the RCP8.5 scenario. a. The 36-year running trends in PWC index [units: Pa (36 year)^{-1}] and IPO index [units: K (36 year)^{-1}] during 1980-2051 obtained from six LEs. Dashed Lines with squares in purple (orange) and blue (red) indicate the ensemble mean IPO (PWC) trends by LEs and the best match members, with error bars representing one standard deviation ensemble member spread. The horizontal axis marks the end year for the 36-year segment. **b.** Histograms (bars) and fitted distribution (lines) of IPO index trends [units: K (36 year)^{-1}] derived from the best match members. The blue (red) bars and fitted curve show the frequency of occurrence [units: %] of the IPO index trends for present (future) climate. Black dashed line shows the observed IPO index trend. The red and blue triangles denote the ensemble mean of the distribution with the corresponding color. **c.** The same as **b.** but for PWC index trends [units: Pa (36 year)^{-1}]. Differences between the future and present distributions for IPO index trends and PWC index trends are significant at 99% confidence level.

Figure R1. The 36-year running trends in PWC index [units: Pa (36 year)⁻¹] and IPO index [units: K (36 year)⁻¹] without the influence of external forcing during 1980-2051 by BM members obtained from **a.** Fig. 4a, **b.** Supplementary Figure 6b, and **c.** Supplementary Figure 6d. Dashed Lines with squares in blue and red indicate the ensemble mean IPO and PWC index trends, with error bars representing one standard deviation ensemble member spread. The horizontal axis marks the end year for the 36-year segment.

(4) Lines 125-128, reference 19:

Note that Wang et al. 2013 demonstrated that the 3yr running mean AMO and IPO indices are uncorrelated with $r=0.07$ from 1958-2011, but both impacts NHSM. Many existing studies however do show AMO impacts on PWC (e.g., Kucharski et al. 2015; Johnson et al. 2020; Kim et al. 2020). Are they examining interannual timescale without filtering, whereas this MS examines “decadal scale” variability with 9yr running mean?

Kucharski et al, 2015: Atlantic forcing of Pacific decadal variability. *Clim Dyn.*

Johnson et al. 2020: Pacific decadal oscillation remotely forced by the equatorial Pacific and the Atlantic Oceans. *Clim Dyn.*

Kim et al. 2020: Pacific Mean-State Control of Atlantic Multidecadal Oscillation–El Niño Relationship. *J. Clim.*

Thank you for your comment. As you mentioned above, the potential impacts of the AMO on PWC on the interannual and decadal timescales has been emphasized in literatures (Dong et al. 2006; Kucharski et al. 2015; Li et al. 2015; Levine et al. 2018; Johnson et al. 2020; Kim et al. 2020). In this study, we also find AMO can modulate the PWC change on the 36-year timescale (Supplementary Fig. 2). The 36-year running correlation between the PWC index and AMO index ranges from -0.91 to 0.78 in the 1021 years MPI-GE pre-industrial control simulation, indicating an unstable relationship between the AMO and PWC. However, we find that the correlation between AMO and PWC trends during 1980-2015 is not statistically significant based on 100 independent samples (Fig. 2f). This indicates that the phase transition of the AMO has limited influence on the PWC during the focused period. The above results are also confirmed by another five LEs (please see our response to comment 2 of Reviewer #3, and Supplementary Fig. 8 below). We have modified the relevant text in the revision (L127-132) about this point.

L127-132: “The observed correlation coefficient between PWC index and AMO index is within the natural variability range (Supplementary Fig. 2e). However, with a large independent sample size, we cannot find a robust coherent relationship between the AMO trend and PWC trend on a 36-year window (Fig. 2f). This indicates that the phase transition of the AMO has limited influence on PWC change during the focused period”.

References

Dong, B., R. T. Sutton, and A. A. Scaife, 2006: Multidecadal modulation of El Niño–Southern Oscillation (ENSO) variance by Atlantic Ocean sea surface temperatures. *Geophysical Research Letters*, **33**.

Kucharski, F., and Coauthors, 2015: Atlantic forcing of Pacific decadal variability. *Climate Dynamics*, **46**, 2337-2351.

Li, X., S.-P. Xie, S. T. Gille, and C. Yoo, 2015: Atlantic-induced pan-tropical climate change over the past three decades. *Nature Climate Change*, **6**, 275-279.

Levine, A. F. Z., D. M. W. Frierson, and M. J. McPhaden, 2018: AMO Forcing of Multidecadal Pacific ITCZ Variability. *Journal of Climate*, **31**, 5749-5764.

Johnson, Z. F., Y. Chikamoto, S. Y. S. Wang, M. J. McPhaden, and T. Mochizuki, 2020: Pacific decadal oscillation remotely forced by the equatorial Pacific and the Atlantic Oceans. *Climate Dynamics*, **55**, 789-811.

Kim, D., S.-K. Lee, H. Lopez, and M. Goes, 2020: Pacific Mean-State Control of Atlantic Multidecadal Oscillation–El Niño Relationship. *Journal of Climate*, **33**, 4273-4291.

Response to Reviewer #2

Review's comments for the paper (NCOMMS-21-10064-T), entitled "A very likely weakening of Pacific Walker Circulation in constrained near-future projections", submitted to Nature Communications

By using both observations and large ensemble simulations from multimodel simulations, this paper investigated the recent enhancement of the Pacific Walker circulation (PWC) and made constrained near-term projections. The analyses suggested that recent enhanced PWC is very likely due to the internal variability associated with a negative trend of the interdecadal Pacific Oscillation (IPO). The constrained projections by IPO phase change suggested a high probability of a weakened PWC in the near future. Results are interesting and the paper is worth of publishing. However, the paper needs some improvements by properly addressing the comments listed below before it can be accepted for publication in Nature Communications.

We would like to express our appreciation to the reviewer for all the constructive comments that helped us to improve the manuscript. According to the reviewer's suggestions, we have thoroughly revised the manuscript. We have explained the relationship between PWC, IPO and the east-west SST gradient over the tropical Pacific to make it more clear to the readers.

Major comments

1. One major concern the reviewer has is the assurance of the study that the interdecadal Pacific Oscillation (IPO) is driving the interdecadal change of Pacific Walker circulation (PWC). The analyses of both observations and model simulations for historical simulations suggest an association of trends between IPO and PWC with strengthened PWC being accompanied with negative trends of IPO. However, there is no evidence to support the authors' assurance that it is IPO change driving PWC. Authors need to give more convincing evidence to support this assurance.

Thanks for your valuable comment. In the tropics, the PWC and ocean are highly coupled systems via the so-called Bjerknes feedback (Bjerknes 1969). Under the framework of Bjerknes feedback, easterlies along the equator pile up warm water to the western Pacific and produce equatorial upwelling to cool the eastern Pacific. The strengthened zonal SST gradient in turn amplifies the easterlies by enhancing the PWC. Hence, the PWC is directly modulated by the east-west SST gradient over the tropical Pacific.

Previous studies have shown evidences that the recent strengthening of the PWC is due to the La Niña-like cooling in the eastern Pacific (Meng et al. 2012; England et al. 2014; McGregor et al. 2014; Ma and Zhou 2016; Kim and Ha 2017), but the cause for this SST cooling trend is an ongoing matter of debate. In this study, we propose that it is IPO that leads the change of PWC by changing the east-west SST gradient over the tropical Pacific. This is because ocean has a longer memory than the atmosphere, which

is the origin of the decadal signals. In addition, previous studies have suggested that the observed change in the PWC is presumably driven by oceanic rather than atmospheric processes through performing AGCM experiments (DiNezio et al. 2009; Meng et al. 2012; Tokinaga et al. 2012). Hence, if IPO switches its phase, the associated east-west contrast in equatorial Pacific SST will change accordingly, regulating the PWC strength through Bjerknes feedback.

References

Bjerknes, J., 1969: Atmospheric teleconnections from the equatorial Pacific. *Mon. Weather Rev.*, **97**, 163-&.

Meng, Q. J., M. Latif, W. Park, N. S. Keenlyside, V. A. Semenov, and T. Martin, 2012: Twentieth century Walker Circulation change: data analysis and model experiments. *Climate Dynamics*, **38**, 1757-1773.

England, M. H., and Coauthors, 2014: Recent intensification of wind-driven circulation in the Pacific and the ongoing warming hiatus. *Nature Climate Change*, **4**, 222-227.

McGregor, S., A. Timmermann, M. F. Stuecker, M. H. England, M. Merrifield, F.-F. Jin, and Y. Chikamoto, 2014: Recent Walker circulation strengthening and Pacific cooling amplified by Atlantic warming. *Nature Climate Change*, **4**, 888-892.

Ma, S., and T. Zhou, 2016: Robust Strengthening and Westward Shift of the Tropical Pacific Walker Circulation during 1979–2012: A Comparison of 7 Sets of Reanalysis Data and 26 CMIP5 Models. *Journal of Climate*, **29**, 3097-3118.

Kim, B.-H., and K.-J. Ha, 2017: Changes in equatorial zonal circulations and precipitation in the context of the global warming and natural modes. *Climate Dynamics*, **51**, 3999-4013.

DiNezio, P. N., A. C. Clement, G. A. Vecchi, B. J. Soden, B. P. Kirtman, and S.-K. Lee, 2009: Climate Response of the Equatorial Pacific to Global Warming. *Journal of Climate*, **22**, 4873-4892.

Tokinaga, H., S. P. Xie, C. Deser, Y. Kosaka, and Y. M. Okumura, 2012: Slowdown of the Walker circulation driven by tropical Indo-Pacific warming. *Nature*, **491**, 439-443.

2. Another important feature from various analyses is the association between changes in PWC and zonal sea surface temperature (SST) gradient in the tropical Pacific with recent strengthened PWC being associated with enhanced zonal SST gradient. There is a wealth of literature about PWC variability and zonal SST gradient variability on interannual to interdecadal timescales. In the paper, it mentioned the role of zonal SST gradient in the tropical Pacific. A question that arises is therefore how IPO, zonal SST gradient, and PWC changes (trends) are related to each other. Is the IPO affecting PWC through its related zonal SST gradient in the tropical Pacific? If it is, this should be made more clearly in the paper.

Thank you for your comment. Given that the strength of the PWC is tightly coupled to the east-west SST gradient over the equatorial Pacific through Bjerknes feedback (Bjerknes 1969), the IPO can affect the PWC via regulating the associated zonal SST gradient. Specifically, a negative IPO phase with cold SST anomalies over the tropical central-eastern Pacific is associated with a strengthened PWC, and the opposite occurs for the IPO positive phase. Hence, if the IPO switches its phase from positive to negative, the associated east-west contrast in equatorial Pacific SST will enhance, leading to a strengthened PWC, and vice versa. As you suggested, we have addressed this point in the revision to make it more clear.

L97-98: “The strength of the PWC is tightly coupled to the east-west SST gradient over the equatorial Pacific through Bjerknes feedback”.

L112-114: “If IPO switches its phase, the associated east-west contrast in equatorial Pacific SST will change hence the PWC strength”.

Reference

Bjerknes, J., 1969: Atmospheric teleconnections from the equatorial Pacific. *Mon. Weather Rev.*, **97**, 163-&.

Minor comments

1. Lines 71-72. See major comment 1.

Please see our response to the reviewer’s comment 1.

2. Lines 78-79. Quantify the forced trend.

Done. Thank you. The forced trend is $9.9 \text{ Pa (36-year)}^{-1}$. We have revised the relevant text in L78-79.

3. Lines 97-122. It is not clear to the reviewer how the authors get this conclusion that IPO regulates PWC variability since all these analyses only demonstrate an association between two phenomena and do not suggest the causality. See major comment 1.

Thank you for your comment. In this study, we have shown the possible trends of PWC under both positive and negative phase of IPO, which is kind of a strong statistic of conditional probability that can be used to understand the causality (Marotzke 2018). More details about the relationship between IPO and PWC in the decadal timescale have been discussed above. Please see our response to the reviewer’s comment 1.

Reference

Marotzke, J., 2018: Quantifying the irreducible uncertainty in near - term climate projections. *Wiley Interdisciplinary Reviews: Climate Change*, **10**, e563.

4. Lines 147-148. What is the relationship between zonal SST gradient and IPO? See major comment 2.

Thank you for your comment. The IPO manifests as a low-frequency El Niño-like pattern, with a warm tropical Pacific during its positive phase, and a cool tropical Pacific during its negative phase. Thus, during the positive phase of IPO, the east-west SST gradient is likely to weaken, and vice versa.

5. Lines 175-178 and Lines 184-187. They give similar information.

Thank you for your comment. We have removed the repetitive sentence in L184-187.

6. Lines 179-180. If the reviewer understood correctly, Frequency of occurrence (FOC) shown in figures 4c and d are based on 13 members (BM). Are these FOCs robust since you used such small ensemble members to construct them?

Thanks for the comment. To reply your concern, we use the Kolmogorov-Smirnov (KS) test to determine whether changes in the FOCs at present and future are significant. The KS test is non-parametric and is widely used to test whether two samples are from the same parent distribution. Results show that distributions of both the projected IPO trends and PWC trends are well-distinguished from that of the historical ones, with P values smaller than 0.01. This indicates the size of ensemble members we choose here is reasonable. Based on your comment, we have mentioned this point in the Method part and relevant figure captions in the revision.

L339-341: “The Kolmogorov-Smirnov nonparametric test is applied to determine whether two probability distributions are well-distinguished”.

7. Lines 198-199. See major comment 1.

Please see our response to comment 1.

8. Lines 244-245. “MPI-GE shows reasonable performance in capturing the recent PWC change”. This statement is confusing. Rephrase.

Thank you. We have rephrased this sentence into “MPI-GE performs well in capturing the recent PWC change” (L262).

9. Line 289. “in for”?

Thank you. We have removed the word “in”.

10. Figures. In some figures. Probability density function(PDF) is used while frequency of occurrence (FOC) is used in others. It seems to the reviewer that they are essentially the same thing and would be better to be consistent.

Done, thank you. We have modified all the relevant figures by showing frequency of occurrence (FOC) to be consistent.

Response to Reviewer #3:

The manuscript “A very likely weakening of Pacific Walker Circulation in constrained near-future projections” attempts to make a projection of the Walker Circulation over roughly the next 30 years. The study uses coupled models, which are known to underestimate the observed recent strengthening of the Walker Circulation. To remedy this underestimation, they statistically adjust the models to the observed trend to assess the relative contributions of internal vs. external forcings. Furthermore, they use the models that vary closely with the observations for near future projections. The topic is interesting and worth pursuing, however, some aspects of this study require further development and clarification.

A large limiting factor of the study is the notion that the models can be adjusted or chosen to better represent the observations, however, this does not address the very real possibility that the models are not representing the physics correctly on these timescales (see citations 10, 11, 12 in manuscript). I do not expect this study to remedy the physics in the models, but some of the issues I have with this study are the result of these model deficiencies.

There are two points that give me reservations in recommending publication of this paper, and a number of subpoints for each. However, I would like to give the authors an attempt at addressing these issues for a possible future submission.

We would like to express our appreciation to the reviewer for the constructive comments that helped us to improve the manuscript. In accordance with the reviewer’s comments and suggestions, we have thoroughly revised the manuscript. We have evaluated models’ performance in simulating the IPO-PWC relationship and clarified the role of external forcing.

1. The method underpinning their result of the likely weakening of the Pacific Walker Circulation does not perform well in their verification analysis. In Line 173 the authors state “[t]he weakening of the PWC from the 1950s to 1980s and the strengthening thereafter are well reproduced”, however Sup. Figs. 5b and 5d do not show a strengthening that is comparable to that observed. The red line in Sup. Fig. 5b does not follow the observed black line, and the observed trend in Sup. Fig. 5d is well outside any modelled trend during 1986-2015. This gives doubt to the results shown in Fig. 4a and Fig.4d, which underpins the statement given in the manuscript title.

Thanks for your comment. In the revision, we show 36-year running trends instead of raw annual mean times series of the IPO and PWC index to highlight the decadal variation. To verify the method of selecting members, we choose BM members based on 1950-1985 to see whether they can reproduce the observed PWC change in 1980-2015. The result shows that the selected BM members for the 1950-1985 period indeed capture the observed strengthening of the PWC with increasing trends, although they simulate a smaller magnitude of the PWC change compared with that in the observation (Supplementary Fig. 5a). The observed PWC trends are within the range of the

ensemble member spread, indicating models' ability to reproduce the observed PWC variation (Supplementary Fig. 5). In addition, Reviewer #1 has a similar concern regarding the method of selecting BM members and provided recommendations. Please kindly refer to our response to comment 1 (Part II: Near-term projection for 2016-2051) of Reviewer #1.

1.1. What is the difference in choosing members that are in the same phase, compared to members that have a similar IPO value at the end of your calibration period? I don't think we know enough about the IPO that we can rely on its current oscillation to continue.

Thanks for your comment. We agree with you that it is difficult to predict the IPO by its past state. In this study, we aim to address the role of IPO phase transition in regulating the PWC strength. Instead of precisely predicting the IPO, results from our study suggest that the recovery from negative IPO phase during 1980-2015 period will influence the future multi-decadal trend in the PWC.

To reply your concern, we compare the results from the selected members that have a similar IPO value to observation (hereinafter referred to as "Method1"), with that show the same phase transition with observation (hereinafter referred to as "Method2") (Figure R2 shown below). Since we mainly focus on the periods in which IPO shifts its phase, members with IPO trend larger than half of the observed value are taken as being the same phase with the observation. Based on Method2, 101 members are selected in total for 1980-2015 period and 47 for 2016-2051 period. As seen in Figure R2, there's no significant difference between these two methods, confirming the robustness of the method of selecting BM members as well as our conclusions.

Figure R2. a. The 36-year running trends in PWC index [units: Pa (36 year)⁻¹] and IPO index [units: K (36 year)⁻¹] during 1950-2015 obtained from six LEs. Dashed Lines with squares in blue (red) and purple (orange) indicate the ensemble mean IPO (PWC) trends by best match members for Method1 and Method2, with error bars representing one standard deviation ensemble member spread. Based on Method2, 101 members are selected in total. **b.** The same as **a.** but for 1980-2051 period. Based on Method2, 47 members are selected in total. The horizontal axis marks the end year for the 36-year segment.

1.2. Line 166: State in the main text how you choose 13 members. Also, which models do these members come from?

Thanks for this comment. Among the 13 BM members, 5 members are from MPI-GE, 2 from CanESM2, 1 from CESM1, 3 from CSIRO, and 2 from GFDL-ESM2M. Following your suggestion, we have explained how we choose the BM member in the main text.

L164-167: “Among the 270 realizations, we select 13 members whose simulated IPO is closest in sign and magnitude to observation, with top 5% significant

correlations between the member-simulated and observed IPO time series”.

2. The adjustment used to make the conclusion that 68% of the observed PWC strengthening is due to the IPO needs to be clarified. The adjustment relies on the relationship between 9-year running mean PWC and IPO from 1980-2015, but this relationship is not shown for the models or observations. Are the models realistic in this relationship? Likewise, how do the models perform in simulating the IPO trends compared to observed? My reservations are that if the models cannot simulate an PWC-IPO relationship or IPO trend of similar magnitude to the observations, can the adjustment made to the models be believed?

Thanks for the comment. The correlation coefficient between the observed IPO and PWC index time series during 1980-2015 is -0.87, within the range from -0.97 to 0.60 among the 270 realizations from six LEs, suggesting that models are able to reproduce this relationship. To reply your concern, we then evaluate the six models' ability in simulating the PWC change during the period of 1980-2015 and its relation to IPO (Supplementary Fig. 7). Although the multimember ensemble mean of most models cannot reproduce the recent strengthening of the PWC, we find some members from the LEs can reasonably capture the observed PWC change and its relationship with IPO (Supplementary Fig. 7), with a comparable magnitude.

As for your second question, we show the relationship between the IPO trends and PWC trends for the six LEs (please see Figs. 2e, f and Supplementary Fig. 8 below). As seen from Figs. 2e, f and Supplementary Fig. 8, all the LEs have the ability to reproduce the observed IPO trend, while the simulated PWC trends are generally weaker than the observation except for MPI-GE. This might be due to the smaller size of samples compared with MPI-GE, or models' deficiencies in the simulation of decadal variability of the PWC strength (Kociuba and Power 2015).

For the last question, in the procedure of the adjustment, we first exclude the original IPO-related PWC trends in each member from the LEs, and then add the observed IPO-related PWC trends back through regression. The adjustment term we add back depends on the difference between the observed and simulated IPO trend. After adjustment, all the LEs members are influenced by the same observational IPO phase transition. This method has been widely used to study the impact of Pacific multidecadal variability on the historical changes of Indian summer monsoon (Salzmann and Cherian 2015; Huang et al. 2020).

Based on your comments, we have added Supplementary Fig. 8, presented the model evaluation results in the Supplementary information, and mentioned this point in the main text.

L266-267: “All the models are capable to simulate the IPO-PWC relationship (Supplementary Figs. 7 and 8)”.

Supplementary information, L114-120: “To evaluate models' ability, we examine

their performance in the simulation of the PWC change and its relation to IPO for the 1980-2015 period (Supplementary Figs. 7 and 8). Although the multimember ensemble mean of most models cannot reproduce the recent strengthening of the PWC, we find some members from the LEs can reasonably capture the observed PWC change and its relationship with IPO, with a comparable magnitude (Supplementary Fig. 7). Moreover, all the LEs are capable to simulate the negative IPO-PWC relationship (Supplementary Fig. 8)”.

Supplementary Figure 7 | Evaluation of LEs in simulating the PWC change during the period of 1980-2015 and its relation to IPO. Spatial distribution of trends in SLP [units: Pa (36 year)⁻¹] derived from **a.** HadSLP2 and **c.** member 46 (NO.46) from MPI-GE, **e.** member 21 (NO.21) from CanESM2, **g.** member 7 (NO.7) from CESM1, **i.** member 27 (NO.27) from GFDL-ESMEM, **k.** member 7 (NO.7) from CSIRO, and **m.**

member 9 (NO.9) from GFDL-CM3. **b.** The regressed SLP [units: Pa] from the HadSLP2 with respect to the standardized IPO index from ERSST v4. **d., f., h., j., l., n.** The same as **b.**, but for NO.46 from MPI-GE, NO.21 from CanESM2, NO.7 from CESM1, NO.27 from GFDL-ESMEM, NO.7 from CSIRO and NO.9 from GFDL-CM3. Slant hatching denotes trends significant at the 95% confidence level.

Supplementary Figure 8 | Scatter plot of IPO index trends [x axis, units: $K (36 \text{ year})^{-1}$] and PWC index trends [y axis, units: $\text{Pa} (36 \text{ year})^{-1}$] during 1980-2015 for **a.** CanESM2, **c.** CESM1, **e.** GFDL-ESM2M, **g.** CSIRO, **i.** GFDL-CM3, separately. Red asterisk denotes the observed value. The correlation coefficient is indicated at the top of the panel. **b., d., f., h., j.** The same as **a., c., e., g., i.**, but for AMO index trends.

References

Kociuba, G., and S. B. Power, 2015: Inability of CMIP5 Models to Simulate Recent

Strengthening of the Walker Circulation: Implications for Projections. *Journal of Climate*, **28**, 20-35.

Salzmann, M., and R. Cherian, 2015: On the enhancement of the Indian summer monsoon drying by Pacific multidecadal variability during the latter half of the twentieth century. *Journal of Geophysical Research: Atmospheres*, **120**, 9103-9118.

Huang, X., and Coauthors, 2020: The Recent Decline and Recovery of Indian Summer Monsoon Rainfall: Relative Roles of External Forcing and Internal Variability. *Journal of Climate*, **33**, 5035-5060.

2.1. In Figure 3a, why are the forced trends different between models? Considering the models have the same forcing, I would assume you would get a consistent response. I don't follow why it is due to the model spread of the equatorial zonal SST gradient mentioned in line 147 and the figures you reference here.

Thanks for making this important point. On the long-term timescale, climate models tend to project weakened PWC under GHG forcing (Vecchi et al. 2006). However, the role of external forcing in the recent strengthening of the PWC remains controversial. This is because the balance between the forced response to aerosol forcing which tends to strengthen the PWC, and GHG forcing which tends to weaken the PWC may not be correct in climate models (DiNezio et al. 2013; Kociuba and Power 2015). Thus, models may overestimate the weakening of the PWC caused by GHG forcing or underestimate the strengthening of the PWC caused by aerosol forcing, leading to large model spread in the PWC change in response to external forcing.

Based on the suggestion from Reviewer #1, we divide six LEs into two groups: Group-S (MPI-GE and GFDL-ESM2M, 130 members in total), in which external forcing induced a strengthened PWC during 1980-2015, and Group-W (CanESM2, CESM, CSIRO and GFDL-CM3, 140 members in total), in which external forcing induced a weakened PWC. The results show that external forcing plays a positive role in the PWC strengthening during the historical period in Group-S, while it tends to weaken the PWC in Group-W (Supplementary Figures 6a and c). More details about this point can be found in our response to comment 2 (Part II: Near-term projection for 2016-2051), Reviewer #1.

The reason we attributed the model spread to the zonal SST gradient over the equatorial is that it is the most important driving force of PWC strength through the Bjerknes feedback (Bjerknes 1969). As seen from Fig. 2b and Supplementary Fig. 3, models (i.e. CanESM2) simulate a weakened PWC under external forcing are accompanied by a weakened east-west SST gradient, while models (i.e. MPI-GE) simulated a strengthened PWC are associated with increased SST gradient.

References

Vecchi, G. A., B. J. Soden, A. T. Wittenberg, I. M. Held, A. Leetmaa, and M. J. Harrison, 2006: Weakening of tropical Pacific atmospheric circulation due to anthropogenic

forcing. *Nature*, **441**, 73-76.

DiNezio, P. N., G. A. Vecchi, and A. C. Clement, 2013: Detectability of changes in the Walker circulation in response to global warming. *J. Clim.*, **26**, 4038–4048.

Kociuba, G., and S. B. Power, 2015: Inability of CMIP5 Models to Simulate Recent Strengthening of the Walker Circulation: Implications for Projections. *Journal of Climate*, **28**, 20-35.

Bjerknes, J., 1969: Atmospheric teleconnections from the equatorial Pacific. *Mon. Weather Rev.*, **97**, 163-&.

2.2. Can you clarify how $\partial PWC_{internal}(i)/\partial t$ is calculated in Equation 7?

The term $\frac{\partial PWC_{internal}(i)}{\partial t}$ in Equation 7 represents the internal component of the PWC trend and can be obtained by the following two steps. Firstly, we remove the multi-member ensemble mean of each LEs to obtain the internal component of the PWC index. Then, we calculate the trend of the internal PWC index to get this term. We have explained the relevant meaning in the revision (L322).

2.3. Line 149: The adjusted PWC trend and observed PWC trend do not seem “close” to me, their means (or medians?) are not in the other’s range. A statistical test would help this disagreement.

Thank you for the question. The point we hope to emphasize here is that the adjusted PWC trends are closer to the observed PWC trends than the original ones. For example, the model-averaged externally forced PWC trend is -5.9 (~-21.0-9.9) Pa (36 year)⁻¹, and it increases to 57.1 (~30.4-79.0) Pa (36 year)⁻¹ after adjustment, much closer to the observed magnitude of 96.5 (~67.5-141.7) Pa (36 year)⁻¹. We have modified the word “close” to “closer” to make it more clear (L149).

2.4. Fig. 2 e and f: To help address my second main point, you could include an observational point, and provide these scatterplots for other models.

Thanks. Added in Supplementary Figure 8 as suggested. Please see our response to comment 2 above.

2.5. Supp. Figure 2e: Provide this plot for the IPO, it would also help address my second main point. If the running correlation of PWC and IPO shows the same variation as shown in this figure, then it would have implications on the adjustment. Along these lines, if there is a relationship between the IPO-PWC relationship and the IPO trend, could you make a more accurate adjustment?

Thank you for sharing your insights with us. Figure R3 shows the 36-year running correlation between the PWC and IPO index for MPI-GE pre-industrial control simulation. Although there are some decadal variations, the PWC is negatively

statistically correlated with the IPO, indicating a more robust relationship of PWC with IPO than the AMO shown in Supplementary Fig. 2e. Specifically, the positive phase of IPO associated with decreased east-west SST gradient leads to a weakened PWC, and vice versa. As described by Equation 3 and Equation 7 in the Method section of the text, the adjustment term is based on the term $r_{PWC,IPO}(i)$, which is the regression coefficient of the 9-year running-mean PWC index regressed onto the IPO index over the same time period. Hence the physical background behind the adjustment procedure is the IPO-PWC relationship as you suggested.

Figure R3. The 36-year running correlation between the PWC index and IPO index. The gray dashed line denotes the observed correlation coefficient ($cc = -0.87$) between the PWC index and IPO index during the period of 1980-2015.

REVIEWER COMMENTS

Reviewer #1 (Remarks to the Author):

The authors have adequately addressed my previous comments and suggestions, and carefully revised the manuscript. I have no further comments to the revised version.

Reviewer #2 (Remarks to the Author):

Review's comments for the revised paper (NCOMMS-21-10064A), entitled "A very likely weakening of Pacific Walker Circulation in constrained near-future projections", submitted to Nature Communications

My comments on the early version of the paper have been addressed in a satisfactory manner in the revised version. The paper is, therefore, acceptable for publication.

Reviewer #3 (Remarks to the Author):

The authors of "A very likely weakening of Pacific Walker Circulation in constrained near-future projections" have attempted to address the numerous concerns of the reviewers, I thank them for their efforts. The authors have performed additional analysis to address many of my concerns, but I feel the new manuscript is still deficient in key areas. The authors have substantially changed Figure 4 of the manuscript in an attempt to bring out the significance of the results, however this new perspective does not reduce my initial concerns but actually raises more. Additionally, many results in the new submission are subtly different for reasons I am unsure, but this also reduces the significance of the results in key areas. As a result of this and other uncertainties, I cannot recommend publication in this journal. Specific arguments of this decision are as follows:

1. The authors could not adequately address the issues I had in my first main point of the first review. That is the proof that the methods underpinning the statement made in the manuscript title performs to the required standard to make an improved projection. The issues stem around the performance of the models and the interpretation of the results shown in Supplementary Fig. 5. Regarding line 181 of the new manuscript, the observed PWC trend is not captured by the model spread at numerous points (e.g. the trends ending in years 1990, 1995, 2000, 2010 and 2015 [Supplementary Fig. 5a]), showing that the models are not capable of simulating the observed PWC trends. Following this point, regarding line 178, I do not agree that the 1980-2015 PWC trend in the BM members are well simulated. Only 7-9 of 13 members can simulate a clear positive PWC trend seen in the observations (Supplementary Figure 5c). These results are weaker than shown in the first submission.

2. The forcing component is not removed from PWC, yet it is in IPO (see Methods). This can be seen in Fig. 4a with LE(PWC) showing a decreasing trend. If the forcing effect (i.e. LE(PWC)) is removed from the BM members(PWC) the trends in BM members(PWC) is likely not significant. This reduces the second half of the manuscript to the results already published in Vecchi and Soden (2007).

3. The representation of the spread in Fig. 4a and Supplementary Fig. 5a has changed to show less information. Previously the full model spread was shown, but now only the one standard deviation

of the spread is shown. More concerning is that no effort has been made to make the one standard deviation bars legible. If they are one standard deviation then the bar's positive and negative length would be equal, but for many this is not the case. Likewise, the one standard deviation bars cannot be seen at all for LEs(PWC), which as noted in my second point, is critical for comparison.

4. The authors introduced the Kolmogorov-Smirnov test in response to reviewers' comments about the significance of the results, but only used it to prove the BM members contain a multidecadal cycle. A more instructive test would be to assess if the BM members perform better than the remaining ensemble for the same trend. E.g., for the 1980-2015 trends in Supplementary Figure 5c, compare trends of the BM members to the trends of all the other members for the 1980-2015 period. Likewise, for Figure 4, which underpins the statement made in the manuscript title.

5. In regard to my second major point made in my first review, thank you for addressing my questions. Both $r(\text{PWC}, \text{IPO})$ and $d\text{IPO}/dt$ show large ranges in the models, including positive and negative values. This does not reassure me as the models can have an IPO-PWC relationship opposite to that observed, suggesting a completely different process. For me to accept this method I would like some reassurance that $r(\text{PWC}, \text{IPO})$ vs. $d\text{IPO}/dt$ shows no relationship. If there is a relationship then you cannot artificially change the slope of $d\text{IPO}/dt$ without changing $r(\text{PWC}, \text{IPO})$.

Response to reviewers' comments of NCOMMS-21-10064A "A very likely weakening of Pacific Walker Circulation in constrained near-future projections"

We wish to express our appreciation to the reviewers for another extremely thorough and detailed review. The comments provided have served to improve the paper for a second time. Below, we have provided point-by-point responses to each of the comments from reviewer#3. In the following, the reviewer's comments are written in black, followed by our response in blue.

Response to Reviewer #1:

The authors have adequately addressed my previous comments and suggestions, and carefully revised the manuscript. I have no further comments to the revised version.

We would like to express our appreciation to the reviewer for all the constructive comments and suggestions that helped us to improve the manuscript.

Response to Reviewer #2:

Review's comments for the revised paper (NCOMMS-21-10064A), entitled "A very likely weakening of Pacific Walker Circulation in constrained near-future projections", submitted to Nature Communications

My comments on the early version of the paper have been addressed in a satisfactorily manner in the revised version. The paper is, therefore, acceptable for publication.

We would like to express our appreciation to the reviewer for all the constructive comments and suggestions that helped us to improve the manuscript.

Response to Reviewer #3:

The authors of "A very likely weakening of Pacific Walker Circulation in constrained near-future projections" have attempted to address the numerous concerns of the reviewers, I thank them for their efforts. The authors have performed additional analysis to address many of my concerns, but I feel the new manuscript is still deficient in key areas. The authors have substantially changed Figure 4 of the manuscript in an attempt to bring out the significance of the results, however this new perspective does not reduce my initial concerns but actually raises more. Additionally, many results in the new submission are subtly different for reasons I am unsure, but this also reduces the significance of the results in key areas. As a result of this and other uncertainties, I cannot recommend publication in this journal. Specific arguments of this decision are as follows:

We would like to express our appreciation to the reviewer for all the constructive comments. In this round of revision, we have checked the observational datasets and

revised the manuscript thoroughly. We hope that our responses now adequately satisfy the reviewer's concerns.

1. The authors could not adequately address the issues I had in my first main point of the first review. That is the proof that the methods underpinning the statement made in the manuscript title performs to the required standard to make an improved projection. The issues stem around the performance of the models and the interpretation of the results shown in Supplementary Fig. 5. Regarding line 181 of the new manuscript, the observed PWC trend is not captured by the model spread at numerous points (e.g. the trends ending in years 1990, 1995, 2000, 2010 and 2015 [Supplementary Fig. 5a]), showing that the models are not capable of simulating the observed PWC trends. Following this point, regarding line 178, I do not agree that the 1980-2015 PWC trend in the BM members are well simulated. Only 7-9 of 13 members can simulate a clear positive PWC trend seen in the observations (Supplementary Figure 5c). These results are weaker than shown in the first submission.

We appreciate the reviewer's comment on this point. As the reviewer mentioned, in the earlier revision, the observed 36-year running trends of the PWC index for 1950-2015 period are not fully captured by the model spread especially for the second half of the twentieth century (Supplementary Fig. 5a in the first revision). In previous versions, we used two observational datasets to examine the PWC decadal change: HadSLP2 and ICOADS. After a further careful examination of the observational data uncertainty and bias, we find that the models with large ensemble can reasonably capture the observed change from HadSLP2, but the large ensemble simulations fail at capturing PWC change at the early period of ICOADS data (please see Figure R1b below). We then investigated the quality of observational data and consulted the ICOADS data team on the quality of SLP data. We find that the ICOADS data set is unreliable for trend analysis of PWC change. This is because ICOADS data set is based on surface marine data and is *in situ* alone with no spatial interpolation, which lacks quality control and has many missing data points (L'Heureux et al. 2013; Freeman et al. 2017). The substantial missing values in the tropics from early period make ICOADS SLP much noisier than other observational data sets. The obvious bias of ICOADS SLP data in representing tropical Walker Circulation have been investigated and documented in previous study of L'Heureux et al.(2013).

To examine the applicability of the ICOADS data set, we compare the time series and 36-year running trends of the PWC index calculated from the raw ICOADS data with its estimate as recommended by L'Heureux et al. (2013) (Figure R1). In the new estimate, only well-sampled grid boxes in ICOADS are used to calculate regional SLP trends, whereas the others have used linear interpolation to infill missing data (Figure R2). The new ICOADS estimate is obtained by averaging together four sub-estimates based on different criteria (please see figure caption of Figure R2 for details). In addition, another two 20th century reanalysis data sets of 20CR and ERA-20C for 1950-2010 are added and two data sets from the satellite era of ERAIM and ERA5 for 1980-

2015 are used to present the observational uncertainty. With the ICOADS new estimate data, we find the PWC trends in early period agree well with other observational data sets and are also captured by models with large ensemble (Figure R1b). This further demonstrates that the inconsistency between model and observational record is mainly caused by the data bias from ICOADS. Therefore, considering the obvious bias of ICOADS SLP data in presenting tropical Walker Circulation, we decide to exclude ICOADS SLP from our study as shown by L’Heureux et al. (2013) (please see Supplementary Fig. 5a below).

To further verify the method of selecting members, we choose BM members based on 1950-1985 to see whether they can reproduce the observed PWC change in 1980-2015. The result shows that although the magnitude of the PWC change simulated by the selected BM members is weaker than observations, the observed strengthening of the PWC with increasing trends are indeed captured, which adds creditability of this method (Supplementary Fig. 5b). In addition, among the selected 13 members we do find some individual realizations match the observational tendency of PWC better than the ensemble mean shown here. This addresses the importance of even larger ensemble size, which would allow us to choose more BM members. Moreover, in this study we mainly focus on the change of probability. Hence, 7-9 of 13 members simulate an observed positive PWC trend, indicating that the PWC is more likely to strengthen instead of weakening for the corresponding period.

As for your last concern, the results in Supplementary Fig. 5 in the first revision are slightly different from that in the first submission. This is because the methods we used to extract the decadal signals in the first submission were not unified, i.e. in some cases we applied a 9-year running mean to raw data to isolate decadal signals, while in other cases we used Lanczos filter. In the first round of revised manuscript, we unified our method and used the 9-year running mean through all the analyses.

Based on above discussion, we have excluded the results of ICOADS and updated Supplementary Fig. 5 and the relevant statement and figures in the revised manuscript.

L26-28: “Models with sufficient ensemble members can reasonably capture the observed PWC and IPO changes”.

L242-245: “To verify the decadal change of the PWC, we also use the time series of SLP derived from the NOAA-CIRES 20th century reanalysis Version 2c (20CR)⁴⁷ and the ECMWF ERA-20C reanalysis⁴⁸ over the period 1950-2010”.

L246-249: “We exclude SLP data of International Comprehensive Ocean-Atmosphere Data Set (ICOADS) from this study, because previous studies have shown obvious bias of ICOADS SLP data in presenting the Pacific Walker circulation⁵¹”.

L523-L525: “We thank Dr. Clara Deser from the National Center for Atmospheric Research in Boulder USA for her very helpful comments and suggestions on observational data quality control”.

Figure R1. a. Time series of 9-year running mean of PWC index [units: Pa] for 1950-2015 derived from LEs(red) along with HadSLP2, ICOADS, ICOADS estimate and reanalyses. Red shading denotes the range across all the ensemble members. **b.** The 36-year running trends in PWC index [units: Pa (36 year)⁻¹] obtained from the LEs and seven datasets in **a.** with corresponding colors. The error bars representing the ensemble member spread. The horizontal axis marks the end year for the 36-year segment. The ICOADS estimate shown is obtained by averaging together four different sub-estimates in Figure R2.

Figure R2. Time series of 9-year running mean of PWC index [units: Pa] for 1950-2015 in ICOADS (black) and four ICOADS estimates. Following L'Heureux et al., four different estimates are computed only for grid boxes that fulfilled certain criteria: 4mth/yr (gray), PWC index estimates in a grid box are required to have at least 4 months of data within each and every year of the entire record; 6mth/yr (green), similar to 4mth/yr but 6 months of data is required; 75% (blue), PWC index estimates in a grid box must have at least 75% of monthly data for the entire record; 95% (red), similar to 75% but for 95% of data.

Supplementary Fig. 5 | Verifying the method of selecting ensemble members according to observed IPO. **a.** The 36-year running trends in PWC index [units: Pa (36 year)⁻¹] and IPO index [units: K (36 year)⁻¹] during 1950-2015 obtained from six LEs. Dashed Lines with squares in purple (orange) and blue (red) indicate the ensemble mean IPO (PWC) index trends by LEs, with error bars representing the ensemble member spread. Black markers are from different observational and reanalysis datasets. **b.** The same as **a.** but by the best match members. **c.** Histograms (bars) and fitted distribution (lines) of IPO index trends [units: K (36 year)⁻¹] derived from the best match members. The blue (red) bars and fitted curve show the frequency of occurrence [units: %] of the IPO trends for the 1950-1985 (1980-2015) period. The blue and red solid lines show the observed IPO index trend for the two periods. The blue and red triangles denote the ensemble mean of the distribution with the corresponding color. **d.** The same as **c.** but for PWC index trends [units: Pa (36 year)⁻¹] along with HadSLP2 and reanalyses. Differences between the distributions for the 1980-2015 and 1950-1985 periods are significant at 99% confidence level.

References:

Freeman, E., and Coauthors, 2017: ICOADS Release 3.0: a major update to the historical marine climate record. *International Journal of Climatology*, **37**, 2211-2232.

L'Heureux, M. L., S. Lee, and B. Lyon, 2013: Recent multidecadal strengthening of the Walker circulation across the tropical Pacific. *Nature Climate Change*, **3**, 571-576.

2. The forcing component is not removed from PWC, yet it is in IPO (see Methods). This can be seen in Fig. 4a with LE(PWC) showing a decreasing trend. If the forcing effect (i.e. LE(PWC)) is removed from the BM members(PWC) the trends in BM members(PWC) is likely not significant. This reduces the second half of the manuscript to the results already published in Vecchi and Soden (2007).

Thanks. As we stated in Page11 Line 224-227 in previous draft of the manuscript, while we agree that external forcing can also affect the PWC change, what we hope to highlight here is that it is internal variability that plays the major role in modulating the present and near-future PWC change. As seen from the revised Fig. 4a, the projected PWC change corresponds well with the IPO phase transition (Fig. 4a). The IPO trends calculated based on the 36-year running window change sign from negative to positive during 2016-2051, with reversed sign change of the PWC trends throughout the corresponding period.

To explore the role of external forcing, we divide six models into two groups: Group-S (MPI-GE and GFDL-ESM2M, 130 members in total), in which external forcing induced a strengthened PWC during 1980-2015, and Group-W (CanESM2, CESM, CSIRO and GFDL-CM3, 140 members in total), in which external forcing induced a weakened PWC. We then compare projected PWC change for future periods (2016-

2051) from those two groups (see Supplementary Fig. 6b and d). The results show that in the near-future, both LE groups project a negative-to-positive IPO phase transition, as well as the weakening of the PWC (Supplementary Figs. 6b and d). As for external forcing, it is likely to amplify the negative PWC trends in both LE groups (Supplementary Figures 6b and d). The above results confirm the robustness of the method of selecting BM members and the conclusion that the PWC is very likely to weaken, which is mainly related to the recovery of IPO in the coming decades.

To further clarify this point, we examine the internal IPO and PWC trends by removing the influence of external forcing (i.e. remove each LE's ensemble mean) for all six LEs, Group-S and Group-W (Figure R3 shown below). The results are consistent with that from Fig. 4 and Supplementary Fig. 6, confirming the robustness of our conclusions.

To quantify the contribution of internal variability and external forcing to the near-term changes of PWC, we first calculate the magnitude of total PWC trends shown in Fig.4a, and also that caused by internal variability shown in Fig. R3a. The external forcing is excluded via removing ensemble mean of trends from all large ensemble members. The variability is then calculated by standard deviation of the temporal evolution of the 36-year running trends. We measure the contribution of internal variability to the total change by the ratio between the two estimated magnitudes. We find that internal variability contributes about 71% to the total magnitude of the 36-year running trends, while external forcing makes up the rest. The corresponding contribution for Group-S (Group-W) is 64% (68%). The estimates indicate the dominant role of internal variability.

Based on above discussion, we have added Supplementary Figure 6 and discussed the role of external forcing in Lines 219-237 in the main text and Lines 85-102 in the Supplementary information.

L219-237: “In addition, the imbalance between the forced response to aerosol forcing (which tends to strengthen the PWC) and GHG forcing (which tends to weaken the PWC) in climate models may obscure the role of external forcing^{12,32}. Although spread exists in the externally-forced response among different models for the past (Fig. 3a), external forcing is likely to amplify the negative PWC trends in the near-future (Fig. 4a and Supplementary Fig. 6). This is because CMIP5 models tend to project an El Niño-like warming under strengthened GHG forcing, with reduced east-west Pacific SST gradient^{31,45}. To quantify the contribution of internal variability and external forcing to the near-term changes of PWC, we first calculate the magnitude of total PWC trends shown in Fig.4a, and also that caused by internal variability. The external forcing is excluded via removing ensemble mean of trends from all large ensemble members. The variability is then calculated by standard deviation of the temporal evolution of the 36-year running trends. We measure the contribution of internal variability to the total change by the ratio between the two estimated magnitudes. We find that internal variability contributes about 71% to the total magnitude of the 36-year running trends, while external

forcing makes up the rest, which indicates the dominant role of internal variability. Hence, the IPO phase reversal from negative to positive superimposes on the forced El Niño-like pattern, leading to the weakening of the PWC in the coming decades”.

Supplementary information, L85-102: “Climate models tend to project weakened PWC under GHG forcing on the long-term timescale¹ but the role of external forcing in recent strengthening of the PWC remains controversial. This is because the balance between the forced response to aerosol forcing which tends to strengthen the PWC, and GHG forcing which tends to weaken the PWC may not be correct in climate models^{2,3}. To explore the role of external forcing, we divide six LEs into two groups: Group-S (MPI-GE and GFDL-ESM2M, 130 members in total), in which external forcing induced a strengthened PWC during 1980-2015, and Group-W (CanESM2, CESM, CSIRO and GFDL-CM3, 140 members in total), in which external forcing induced a weakened PWC. We then compare the simulated and projected PWC change for both the 1950-2015 and 1980-2051 periods from those two groups (Supplementary Fig. 6). The result shows that external forcing is likely to strengthen the PWC in Group-S over the past several decades, while it tends to weaken the PWC in Group-W (Supplementary Figures 6a and c). In the near-future, however, external forcing is likely to amplify the negative PWC trends in both LE groups (Supplementary Figures 6b and d). The internal variability contributes roughly 64% in Group-S and 68% in Group-W to the total magnitude of the averaged 36-year running trends in the projection. Consequently, internal variability superimposes on the forced response and dominates the PWC change”.

Fig. 4. Effect of IPO on the near-term projection of the PWC under the RCP8.5 scenario. a. The 36-year running trends in PWC index [units: Pa (36 year)⁻¹] and IPO index [units: K (36 year)⁻¹] during 1980-2051 obtained from six LEs. Dashed Lines with squares in purple (orange) and blue (red) indicate the ensemble mean IPO (PWC) index trends by LEs and the best match members, with error bars representing the ensemble member spread. The horizontal axis marks the end year for the 36-year segment. **b.** Histograms (bars) and fitted distribution (lines) of IPO index trends [units: K (36 year)⁻¹] derived from the best match members. The blue (red) bars and fitted curve show the frequency of occurrence [units: %] of the IPO index trends for present (future) climate. Black line shows the observed IPO index trend. The red and blue triangles denote the ensemble mean of the distribution with the corresponding color. **c.** The same as **b.** but for PWC index trends [units: Pa (36 year)⁻¹] along with HadSLP2 and reanalyses. Differences between the future and present distributions for IPO index trends and PWC index trends are significant at 99% confidence level.

Supplementary Figure 6 | Role of external forcing. The 36-year running trends in PWC index [units: Pa (36 year)^{-1}] and IPO index [units: K (36 year)^{-1}] during (left) 1950-2015 and (right) 1980-2051 obtained from **a-b.** MPI-GE and GFDL-ESM2M and **c-d.** CanESM2, CESM, CSIRO and GFDL-CM3. Dashed Lines with squares in purple (orange) and blue (red) indicate the ensemble mean IPO (PWC) index trends by LEs and the best match members, with error bars representing the ensemble member spread. The horizontal axis marks the end year for the 36-year segment.

Figure R3. The 36-year running trends in PWC index [units: Pa (36 year)⁻¹] and IPO index [units: K (36 year)⁻¹] without the influence of external forcing during 1980-2051 by BM members obtained from **a.** Fig. 4a, **b.** Supplementary Figure 6b, and **c.** Supplementary Figure 6d. Dashed Lines with squares in blue and red indicate the ensemble mean IPO and PWC index trends, with error bars representing one standard deviation ensemble member spread. The horizontal axis marks the end year for the 36-year segment.

3. The representation of the spread in Fig. 4a and Supplementary Fig. 5a has changed to show less information. Previously the full model spread was shown, but now only the one standard deviation of the spread is shown. More concerning is that no effort has been made to make the one standard deviation bars legible. If they are one standard deviation then the bar's positive and negative length would be equal, but for many this is not the case. Likewise, the one standard deviation bars cannot be seen at all for LEs(PWC), which as noted in my second point, is critical for comparison.

Thank you for raising this point. We apologize that our figures in the first round of revision are not clear enough and somewhat misleading. Actually, in the previous round

of revision, the error bars in Supplementary Fig. 5a show one standard deviation of ensemble member spread, while the shadings show the full member spread. Some of the one standard deviation bars cannot be seen because of the overlap. Based on your comment, we have modified Fig. 4 and Supplementary Fig. 5 and unified the error bars to show the full ensemble spread. Please see the revised Supplementary Fig. 5 in response to your comment 1 and Fig. 4 in response to your comment 2.

4. The authors introduced the Kolmogorov-Smirnov test in response to reviewers' comments about the significance of the results, but only used it to prove the BM members contain a multidecadal cycle. A more instructive test would be to assess if the BM members perform better than the remaining ensemble for the same trend. E.g., for the 1980-2015 trends in Supplementary Figure 5c, compare trends of the BM members to the trends of all the other members for the 1980-2015 period. Likewise, for Figure 4, which underpins the statement made in the manuscript title.

Thank you for this comment. Based on your suggestion, we compare trends by the BM members with that by all the other members for both the 1950-2015 and 1980-2015 periods. The result shows that the BM members do perform better than the remaining ensemble members for the same trend (please see Figure R4 below), confirming the robustness of the method of selecting BM members. Based on your comment, we have added a brief sentence to address this point.

L172-175: "We find that the selected ensemble members which are just in phase with the observed IPO in 1950-1985 do capture the IPO phase shift in 1980-2015 (Supplementary Fig. 5a) and performs better than the remaining ensemble members".

Figure R4. The 36-year running trends in PWC index [units: Pa (36 year)^{-1}] and IPO index [units: K (36 year)^{-1}] during **a.** 1950-2015 and **b.** 1980-2051. Dashed Lines with squares in purple (orange) and blue (red) indicate the ensemble mean IPO (PWC) trends by all the other members and the best match members, with error bars representing the ensemble member spread. The horizontal axis marks the end year for the 36-year segment.

5. In regard to my second major point made in my first review, thank you for addressing my questions. Both $r(\text{PWC}, \text{IPO})$ and $d\text{IPO}/dt$ show large ranges in the models, including positive and negative values. This does not reassure me as the models can have an IPO-PWC relationship opposite to that observed, suggesting a completely different process. For me to accept this method I would like some reassurance that $r(\text{PWC}, \text{IPO})$ vs. $d\text{IPO}/dt$ shows no relationship. If there is a relationship then you cannot artificially change the slope of $d\text{IPO}/dt$ without changing $r(\text{PWC}, \text{IPO})$.

Thank you for this comment. To reply your concern, we examine the inter-member relationship between the $r(\text{PWC}, \text{IPO})$ and the IPO trend (i.e. $d\text{IPO}/dt$) using the MPI-

GE. As seen from Figure R5, there is very low correlation between the IPO trend and $r(\text{PWC}, \text{IPO})$, indicating the feasibility of the adjustment method.

Figure R5. Scatter plot of IPO index trends [x axis, units: $\text{K} (36 \text{ year})^{-1}$] and the correlation coefficient between PWC index and IPO index [y axis] during 1980-2015 for MPI-GE. Red asterisk denotes the observed value. The inter-member correlation coefficient is indicated at the top of the panel.

REVIEWERS' COMMENTS

Reviewer #3 (Remarks to the Author):

The authors performed admirably in addressing many of my concerns. The inclusion of other observational datasets helps to place the model results in a better context, albeit with reduced certainty about observed PWC change. The additional analysis, such as removing the forced component of the PWC before calculations, shows largely consistent results with their original hypothesis. The authors also addressed a concern I had in the adjustment method, finding it is not an issue. I feel the manuscript is now worthy of wider scientific discussion and therefore recommend its publication.

Response to reviewers' comments of NCOMMS-21-10064B "A very likely weakening of Pacific Walker Circulation in constrained near-future projections"

We wish to express our appreciation to the reviewers for all the constructive comments that helped us to improve the manuscript. In the point-by-point response, the reviewer's comments are written in black, followed by our response in blue.

Response to Reviewer #3:

The authors performed admirably in addressing many of my concerns. The inclusion of other observational datasets helps to place the model results in a better context, albeit with reduced certainty about observed PWC change. The additional analysis, such as removing the forced component of the PWC before calculations, shows largely consistent results with their original hypothesis. The authors also addressed a concern I had in the adjustment method, finding it is not an issue. I feel the manuscript is now worthy of wider scientific discussion and therefore recommend its publication.

Thank you very much for your approval for our revision.